# Magnesium modulates phospholipid metabolism to promote bacterial phenotypic resistance to antibiotics

**Hui Li[1,2†], Jun Yang[1,2†], Su-fang Kuang[1], Huan-zhe Fu[1], Hui-yin Lin[1], Bo Peng[1,2*]**

[1]State Key Laboratory of Biocontrol, School of Life Sciences, Guangdong Province Key Laboratory for Pharmaceutical Functional Genes, Southern Marine Science and Engineering Guangdong Laboratory (Zhuhai), Sun Yat-sen University, Guangzhou, China; [2]Laboratory for Marine Biology and Biotechnology, Marine Fisheries Science and Food Production Processes, Qingdao Marine Science and Technology Center, Qingdao, China

## eLife Assessment

The study explored the influence of magnesium on phenotypic antibiotic resistance in two strains of *Vibrios*: *V. alginolyticus* ATCC33787 and *V. parahaemolyticus* VP01. This research is **fundamental** for revealing the phenotypic antibiotic resistance mechanism utilized by the specified model bacteria in elevated levels of magnesium. The study produced **convincing** evidence indicating that in high concentrations of magnesium, the efficacy of selected antibiotics was diminished due to decreased biosynthesis of unsaturated fatty acids and PE, along with an increase in the biosynthesis of PG.

***For correspondence:**
pengb26@mail.sysu.edu.cn

[†]These authors contributed equally to this work

**Competing interest:** The authors declare that no competing interests exist.

**Abstract** Non-inheritable antibiotic or phenotypic resistance ensures bacterial survival during antibiotic treatment. However, exogenous factors promoting phenotypic resistance are poorly defined. Here, we demonstrate that *Vibrio alginolyticus* are recalcitrant to killing by a broad spectrum of antibiotics under high magnesium. Functional metabolomics demonstrated that magnesium modulates fatty acid biosynthesis by increasing saturated fatty acid biosynthesis while decreasing unsaturated fatty acid production. Exogenous supplementation of unsaturated and saturated fatty acids increased and decreased bacterial susceptibility to antibiotics, respectively, confirming the role of fatty acids in antibiotic resistance. Functional lipidomics revealed that glycerophospholipid metabolism is the major metabolic pathway remodeled by magnesium, where phosphatidylethanolamine biosynthesis is reduced and phosphatidylglycerol production is increased. This process alters membrane composition, increasing membrane polarization, and decreasing permeability and fluidity, thereby reducing antibiotic uptake by *V. alginolyticus*. These findings suggest the presence of a previously unrecognized metabolic mechanism by which bacteria escape antibiotic killing through the use of an environmental factor.

## Introduction

Non-inheritable antibiotic or phenotypic resistance represents a serious challenge for treating bacterial infections. Phenotypic resistance does not involve genetic mutations and is transient, allowing bacteria to resume normal growth. Biofilm and bacterial persisters are two phenotypic resistance types that have been extensively studied (***Brandis et al., 2023***; ***Corona and Martinez, 2013***). Biofilms have complex structures, containing elements that impede antibiotic diffusion, sequestering and inhibiting their activity (***Corona and Martinez, 2013***). Biofilm-forming bacteria and persisters also

have distinct metabolic states that significantly reduce their antibiotic susceptibility (*Yan and Bassler, 2019*). These two types of phenotypic resistance share the common feature in their retarded or even cease of growth in the presence of antibiotics (*Corona and Martinez, 2013*). However, specific factors that promote phenotypic resistance and allow bacteria to proliferate in the presence of antibiotics remain poorly defined.

Metal ions have a diverse impact on the chemical, physical, and physiological processes of antibiotic resistance (*Booth et al., 2011*; *Poole, 2017*). This includes genetic elements that confer resistance to metals and antibiotics (*Poole, 2017*) and metal cations that directly hinder (or enhance) the activity of specific antibiotic drugs (*Zhang et al., 2014*). The metabolic environment can also impact the sensitivity of bacteria to antibiotics (*Jiang et al., 2023b*; *Lee and Collins, 2011*; *Peng et al., 2015*; *Jiang et al., 2020*; *Zhao et al., 2021*). Light metal ions, such as magnesium, sodium, and potassium, can behave as cofactors for different enzymes and influence drug efficacy. Heavy metal ions, including $Cu^{2+}$ and $Zn^{2+}$, confer resistance to antibiotics (*Yazdankhah et al., 2014*; *Zhang et al., 2019*). Recent reports suggest that sodium negatively regulates redox states to promote the antibiotic resistance of *Vibrio alginolyticus* (*Yang et al., 2018*), while actively growing *Bacillus subtilis* cope with ribosome-targeting antibiotics by modulating ion flux (*Lee et al., 2019*). In Gram-negative bacteria, by contrast, zinc enhances antibiotic efficacy by potentiating carbapenem, fluoroquinolone, and β-lactam-mediated killing (*Isaei et al., 2016*; *Zhang et al., 2014*). Magnesium influences bacterial structure, cell motility, enzyme function, cell signaling, and pathogenesis (*Wang et al., 2019*). This mineral also modulates microbiota to harvest energy from the diet (*García-Legorreta et al., 2020*), allowing *B. subtilis* to cope with ribosome-targeting antibiotics by modulating ion flux (*Lee et al., 2019*). However, the role of magnesium in promoting phenotypic resistance is less well understood.

*Vibrios* inhabit seawater, estuaries, bays, and coastal waters, regions full of metal ions such as magnesium (*Kumarage et al., 2022*). Magnesium is the second most dissolved element in seawater after sodium. At a salinity of 3.5% seawater, the magnesium concentration is about 54 mM (*Tsunogai et al., 1968*), and in deep seawater, can be as high as 2500 mM (*Wang et al., 2024*). *Vibrio parahaemolyticus* and *V. alginolyticus* are two representative *Vibrio* pathogens that infect humans and aquatic animals, resulting in illness and economic loss, respectively (*Grimes, 2020*). (Fluoro)quinolones such as balofloxacin (BLFX) are used to treat *Vibrio* infection; however, resistance has emerged due to overuse (*Suyamud et al., 2024*). Indeed, (fluoro)quinolones are one of China's two primary residual chemicals associated with aquaculture (*Liu et al., 2017*). *Vibrio* can develop quinolone resistance through mutations in the DNA gyrase gene or through plasmid-mediated mechanisms (*Dutta et al., 2021*). Thus, the use of *V. parahaemolyticus* and *V. alginolyticus* as bacterial representatives, and BLFX as a quinolone-based antibacterial representative, can help define novel magnesium-dependent phenotypic resistance mechanisms of pathogenic *Vibrio* species.

The current study evaluated whether magnesium induces phenotypic resistance in *Vibrio* species and defined the molecular/genetic basis for this resistance. Genetic approaches, GC-MS analysis of metabolite and membrane remodeling upon antibiotic exposure, membrane physiology, and extensive antimicrobial susceptibility testing were used for the evaluations.

## Results

### $Mg^{2+}$ promotes phenotypic resistance to antibiotics

Marine environments and agriculture, where antibiotics are commonly used, are rich in magnesium. To investigate whether this mineral impacts antibiotic activity, the minimal inhibitory concentration (MIC) of *V. alginolyticus* ATCC33787 and *V. parahaemolyticus* VP01, which we referred to as ATCC33787 and VP01, isolated from marine aquaculture, to BLFX in Luria–Bertani medium (LB medium) plus 3% NaCl as LBS medium and 'artificial seawater' (ASWT) medium that included the major ion species in marine water (*Wilson, 1975*) (LB medium plus 210 mM NaCl, 35 mM $Mg_2SO_4$, 7 mM KCl, and 7 mM $CaCl_2$) were assessed (*Supplementary file 1a*). The MICs of ATCC33787 to BLFX were 54 and 1 μg/mL in ASWT and LBS medium, respectively. Similarly, the MICs to VP01 were 25 and 3 μg/mL in ASWT and LBS medium, respectively (*Figure 1A*). The role of exogenous NaCl in antibiotic resistance was investigated previously (*Yang et al., 2018*). The current study found that the MIC for BLFX was the same in LB and M9 minimal medium (M9 medium) plus 7 mM KCl or 7 mM $CaCl_2$ (*Figure 1—figure supplement 1*). However, the MIC for BLFX was higher in ASWT medium supplemented with $Mg_2SO_4$

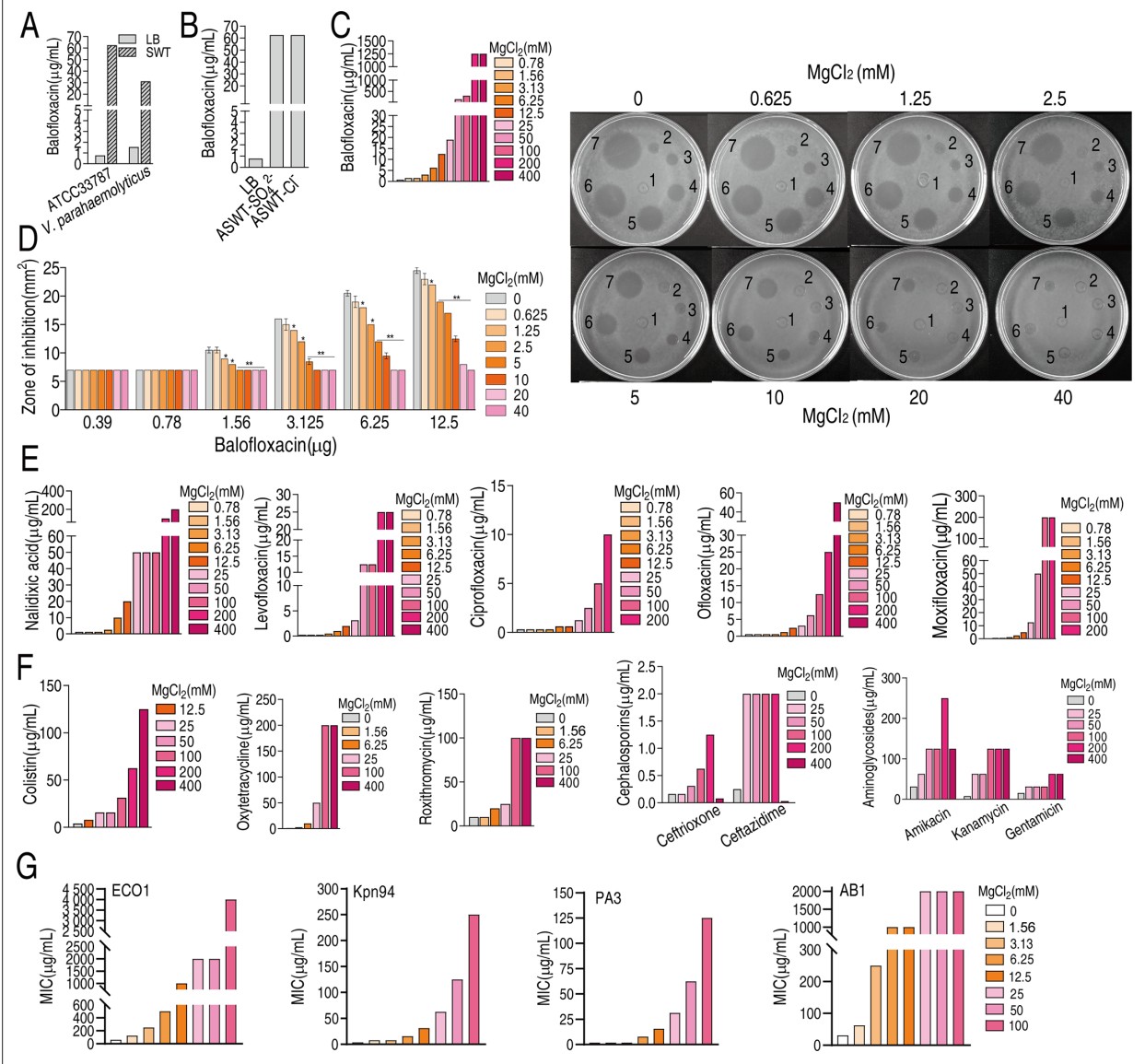

**Figure 1.** Magnesium promotes bacterial resistance to antibiotics. (**A**) Minimal inhibitory concentration (MIC) of ATCC33787 and *V. parahaemolyticus* to balofloxacin (BLFX) in artificial seawater (ASWT) or Luria–Bertani (LB) medium as determined using the microtiter-dilution method. (**B**) MIC of ATCC33787 to BLFX in ASWT with additional MgSO₄ or MgCl₂ as determined using the microtiter-dilution method. (**C**) MIC of ATCC33787 to BLFX in ASWT at the indicated concentrations of MgCl₂ as determined using the microtiter-dilution method. (**D**) MIC of ATCC33787 to BLFX at the indicated concentrations of BLFX and MgCl₂ as determined by the Oxford cup test. The numbers 1, 2, 3, 4, 5, 6, and 7 represent 0, 0.39, 0.78, 1.56, 3.125, 6.25, and 12.5 µg BLFX, respectively. (**E**) MIC of ATCC33787 to other quinolones in ASWT at the indicated concentrations of MgCl₂ as determined using the microtiter-dilution method. (**F**) MIC of ATCC 33787 to other classes of antibiotics in ASWT at the indicated concentrations of MgCl₂ as determined using the microtiter-dilution method. (**G**) MIC of carbapenem-resistant *Escherichia coli*, carbapenem-resistant *Klebsiella pneumoniae*, carbapenem-resistant *Pseudomonas aeruginosa*, and carbapenem-resistant *Acinetobacter baumannii* isolates to BFLX at the indicated concentrations of MgCl₂.

The online version of this article includes the following figure supplement(s) for figure 1:

**Figure supplement 1.** Minimal inhibitory concentration (MIC) of ATCC33787 to balofloxacin (BLFX) in the absence or presence of the indicated concentrations of KCl or CaCl₂ in M9 media.

or MgCl₂ than in LB medium (**Figure 1B**). Mg₂SO₄ or MgCl₂ had no difference on MIC, suggesting it is Mg²⁺ not other ions contribute to the MIC change. The MIC for BLFX increased at higher concentrations of MgCl₂ in ASWT medium. Specifically, adding 50 mM or 200 mM MgCl₂ increased the MIC for BLFX by 200- or 1600-fold, respectively (**Figure 1C**). At BLFX doses of 1.56, 3.125, 6.25, and 12.5 µg, the zone of inhibition decreased with increasing MgCl₂ (**Figure 1D**). Exogenous MgCl₂ also increased

the MICs for other quinolone (e.g., nalidixic acid, levofloxacin, ciprofloxacin, ofloxacin, and moxifloxacin) (*Figure 1E*) and non-quinolone antibiotics including antibacterial peptides (colistin), macrolides (roxithromycin), tetracyclines (oxytetraycline), β-lactams (ceftriaxone, ceftazidime), and aminoglycosides (amikacin, kanamycin, and gentamicin) (*Figure 1F*). Notably, magnesium had a reduced effect on ceftriaxone and gentamicin than other antibiotics. Importantly, exogenous $MgCl_2$ also increased MICs of clinic isolates, carbapenem-resistant *Escherichia coli*, carbapenem-resistant *Klebsiella pneumoniae*, carbapenem-resistant *Pseudomonas aeruginosa*, and carbapenem-resistant *Acinetobacter baumannii* to BLFX (*Figure 1G*). These findings indicate that $Mg^{2+}$ promotes phenotypic resistance.

## $MgCl_2$ affects bacterial metabolism

To investigate how magnesium regulates phenotypic resistance, other effects of magnesium, including its ability to quench BLFX activity and induce efflux pump expression and LPS biosynthesis, were excluded (see Appendix 1). While $Mg^{2+}$ is a co-factor for enzymes (*Garfinkel and Garfinkel, 1985*), whether this mineral can promote antibiotic resistance independent of this function remains unknown.

To better understand how magnesium affects bacterial metabolism, *V. alginolyticus* was cultured in M9 medium in the presence of various amounts of $MgCl_2$ (0, 0.78, 3.125, 12.5, 50, or 200 mM) and the levels of 54 metabolites were assessed by GC-MS. Five biological and two technical replicates were evaluated for each treatment (*Figure 2—figure supplement 1*). The levels of 41 metabolites were differential (p<0.05) and are presented as Z-values (*Figure 2A*, *Figure 2—figure supplement 2*). Orthogonal partial least square discriminant analysis (OPLS-DA) was conducted that separated the six treatments into three groups: group 1 included 0, 0.78, and 3.125 mM $MgCl_2$; group 2 was a singlet of 12.5 mM; and group 3 included 50 and 200 mM $MgCl_2$. Component t[1] distinguishes group 1 from groups 2 and 3; component t[2] distinguishes group 3 from groups 1 and 2 (*Figure 2B*). Discriminating variables are shown as S-plot, where cut-off values are ≥0.05 absolute value of covariance p and 0.5 for correlation p(corr). Six biomarkers/metabolites were selected from component t[1] and p[1]. More specifically, the abundance of cadaverine, urea, palmitic acid, aminoethanol, and fumaric acid were increased, but pyroglutamic acid and glutamic acid were decreased (*Figure 2C*, *Figure 2—figure supplement 3*). Pathway enrichment analysis suggests that 12 pathways are involved. Notably, one of the pathways is biosynthesis of unsaturated fatty acids (terminology in the software; actually, it is biosynthesis of fatty acids) (*Figure 2D*), where the abundance of palmitic acid and stearic acid were increased in an $Mg^{2+}$ dose-dependent manner (*Figure 2E*, *Figure 2—figure supplement 4*). Moreover, palmitic acid was a crucial biomarker (*Figure 2—figure supplement 4*). The increase in fatty acid biosynthesis could be partially explained by an imbalanced pyruvate cycle/TCA cycle, in which fumarate levels increased at higher $Mg^{2+}$ while succinate levels increased at lower $Mg^{2+}$ (*Figure 2—figure supplement 3B*). These findings indicated that glycolysis fluxes into fatty acid biosynthesis rather than the pyruvate cycle/TCA cycle. The relevance of fatty acids and BLFX was demonstrated by the observation that exogenous palmitic acid increased bacterial resistance to BLFX (*Figure 2F*). These results suggest that fatty acid metabolism may be critical to magnesium-based phenotypic resistance.

## $Mg^{2+}$ regulates fatty acid biosynthesis

Acetyl-CoA carboxylase (ACC) catalyzes the conversion of acetyl-CoA to malonyl-CoA, the first step of fatty acid synthesis. Fatty acid biosynthesis, which includes saturated and unsaturated forms, shares common biosynthetic pathways with acetyl-CoA to enoyl-ACP, and is sequentially mediated by *fabD* or *fabH*, *fabB/F*, *fabG*, and *fabA/Z*. The enzyme, enoyl-ACP, produces unsaturated fatty acids or is metabolized to acyl-ACP via *fabV* to produce saturated fatty acids. Unsaturated fatty acids produce saturated fatty acids via *tesA/B* and *yciF* (*Figure 3A*). qRT-PCR was used to quantify the expression of involved genes in bacteria treated with different concentrations of $MgCl_2$. The expression of 18 genes increased following treatment with 50 and/or 200 mM $MgCl_2$ (*Figure 3B*). The expression of *tesA*, *tesB*, and *yciA*, which convert unsaturated to saturated fatty acids, increased at 50 or/and 200 mM $MgCl_2$ (*Figure 3C*). FabA and FabF play key roles in unsaturated fatty acid and fatty acid biosynthesis, respectively (*Feng and Cronan, 2010*; *Lee et al., 2019*). Western blot analysis showed that FabA levels declined and FabF levels increased at higher $MgCl_2$ concentrations (*Figure 3D*). Oxazole-2-amine and triclosan inhibit the synthesis of fatty acids and decrease bacterial survival in the presence of BLFX (*Figure 3E and F*). These results indicate that $Mg^{2+}$ inhibits the synthesis of unsaturated fatty acids and promotes the synthesis of saturated fatty acids.

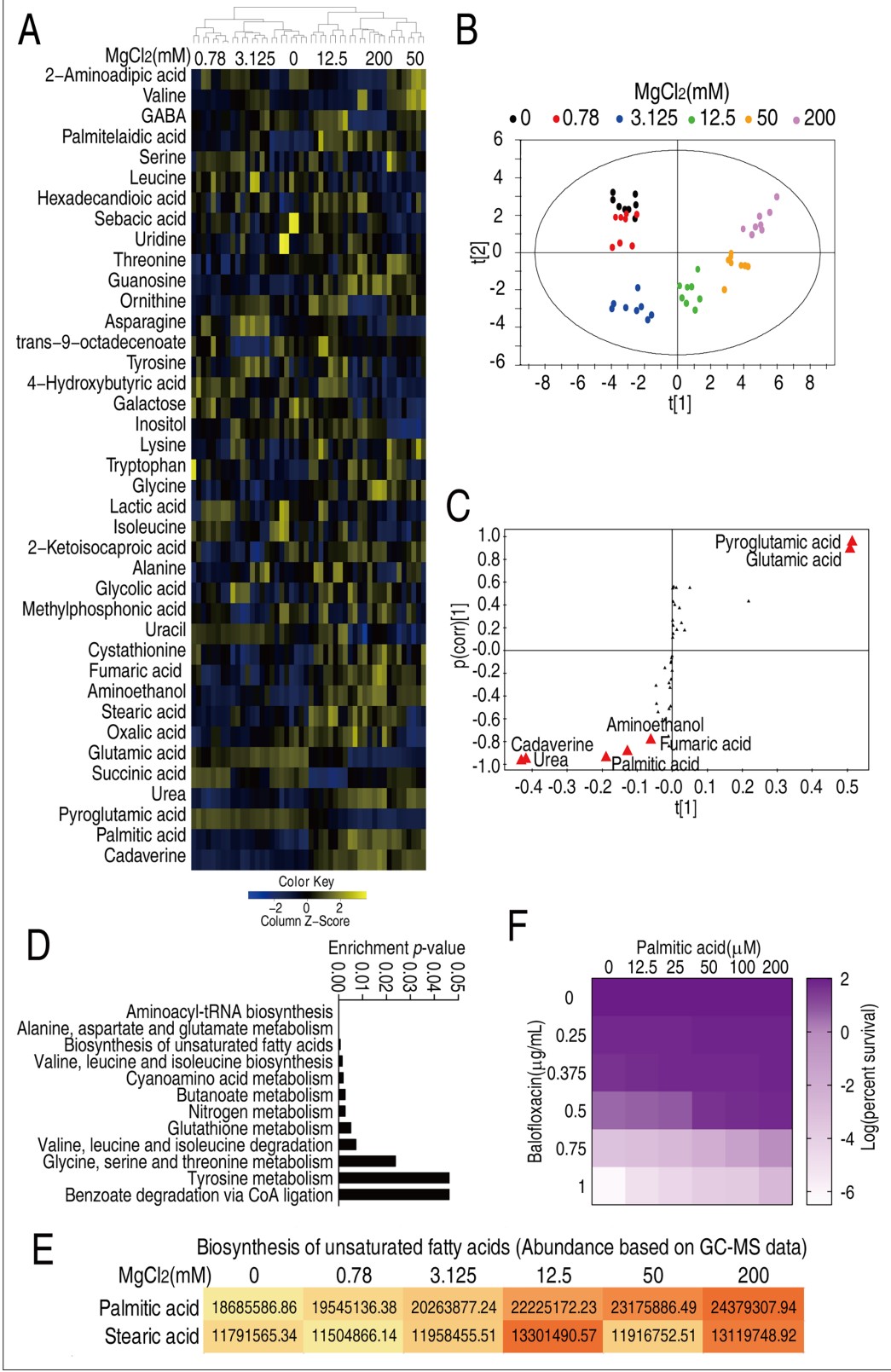

**Figure 2.** Mg²⁺-induced metabolomic change. (**A**) Differential metabolomes in the absence or presence of the indicated concentrations of MgCl₂. The yellow and blue colors indicate an increase or decrease in metabolite levels relative to the median metabolite level, respectively (see color scale). Euclidean distance calculations were used to generate a heatmap that shows clustering of the biological and technical replicates of each treatment.

*Figure 2 continued on next page*

*Figure 2 continued*

(**B**) Orthogonal partial least square discriminant analysis (OPLS-DA) of different $MgCl_2$-induced metabolome concentrations. Each dot represents a technical replicate of samples in the plot. (**C**) S-plot generated from OPLS-DA. Predictive component p[1] and correlation p(corr)[1] differentiate 0, 0.78, and 3.125 mM $MgCl_2$ from 12.5, 50, and 200 mM $MgCl_2$. Predictive component p[2] and correlation p(corr)[2] separate 0, 0.78, 50, and 200 mM $MgCl_2$ from 3.125 and 12.5 mM $MgCl_2$. The triangle represents metabolites in which candidate biomarkers are marked. (**D**) Enriched pathways by differential abundances of metabolites. (**E**) Areas of the peaks of palmitic acid and stearic acid generated by GC-MS analysis. (**F**) Synergy analysis for balofloxacin (BLFX) with palmitic acid for *V. alginolyticus*. Synergy was performed by comparing the dose needed for 50% inhibition of the synergistic agents (white) and non-synergistic (i.e., additive) agents (purple).

The online version of this article includes the following figure supplement(s) for figure 2:

**Figure supplement 1.** Metabolic profiles of *V. alginolyticus* in different concentrations of $MgCl_2$.

**Figure supplement 2.** Heatmap and Z score plots of differential metabolites.

**Figure supplement 3.** Pathway enrichment of differential metabolites.

**Figure supplement 4.** Scatter plots of differential metabolites identified by S-plot.

In addition, we also quantified gene expression during fatty acid degradation to determine whether $Mg^{2+}$ affects this process (*Figure 3G and H*). Interestingly, the expression of genes involved in unsaturated fatty acid degradation decreased in an $MgCl_2$ dose-dependent manner (*Figure 3H*). Western blot analysis confirmed that FadL levels decreased with increasing $MgCl_2$ (*Figure 3I*). FadH degrades unsaturated fatty acids and influences the ratio of unsaturated to saturated fatty acids. Together, these results suggest that $MgCl_2$ inhibits fatty acid degradation.

FabR, FadR, ArcA, and cAMP/CRP are transcriptional factors involved in regulating fatty acid biosynthesis. FabR inhibits unsaturated fatty acid biosynthesis, FadR promotes fatty acid biosynthesis and inhibits degradation, and ArcA and cAMP/CRP inhibit and promote fatty acid degradation, respectively (*Feng and Cronan, 2010*; *Fujita et al., 2007*). The expression of *fabR* and *arcA* increased, while the expression of *fadR* decreased. The expression of N646_1885, which encodes CRP, increased only in the presence of 200 mM $MgCl_2$ (*Figure 3J*). Western blot analysis confirmed that FadR levels declined with increasing $MgCl_2$ concentration (*Figure 3K*). Viability of Δ*fadR* and Δ*arcA* was reduced in the presence of $MgCl_2$, BLFX, or both in a dose-dependent manner (*Figure 3L*). Thus, FadR and ArcA may play roles in bacterial resistance to BLFX, presumably by a mechanism dependent on the abundance of saturated/unsaturated fatty acids.

To investigate whether $Mg^{2+}$ functions through FadR and ArcA, expression of eight fatty acid biosynthetic genes (*accA, accC, fabD, fabH, fabR, fabG, fabV,* and *yciA*) and five fatty acid degradation genes (*fadL, fadD, fadB, fadA,* and *fadH*) were quantified in wildtype, Δ*fadR* or/and Δ*arcA* bacterial strains. As expected, fatty acid biosynthetic gene expression was reduced in Δ*fadR* cells (*Figure 3M*). While higher expression of *fadD, fadB,* and *fadA* (elevated *fadA* only for Δ*fadR* without 200 mM $MgCl_2$) was detected in Δ*fadR* or Δ*acrA* strains, the expression of these genes was negatively regulated by 200 mM $MgCl_2$ (*Figure 3N and O*). These results indicate that $Mg^{2+}$ promotes fatty acid biosynthesis by positively regulating *fadR* and inhibits the degradation of fatty acids through *fadR* and *arcA*.

## $MgCl_2$ affects 16-carbon and 18-carbon fatty acid metabolism

To investigate the effect of $Mg^{2+}$ on the biosynthesis of saturated and unsaturated fatty acids, LC-MS was used to quantify the levels of 16-carbon and 18-carbon fatty acids, the main precursors of lipid biosynthesis (*Zhang and Rock, 2008b*). The abundance of the saturated fatty acid, palmitic acid (C16:0), was higher while the abundance of five unsaturated fatty acids, palmitoleic acid (C16:1), linoelaidic acid (C18:2), linoleic acid (C18:2), alpha-linoleic acid (C18:3), and stearidonic acid (C18:4), was reduced with increasing $MgCl_2$ concentration (*Figure 4A*). The levels of stearic acid (C16:0) and vaccenic acid (C18:1) remained unaffected. Total saturated fatty acid levels increased and total unsaturated fatty acid levels decreased with increasing $Mg^{2+}$ (*Figure 4B*). These results suggest that $Mg^{2+}$ upregulates saturated fatty acid biosynthesis but downregulates unsaturated fatty acid biosynthesis.

Direct exposure to palmitic acid also inhibited BLFX-mediated killing (*Figure 2F*) while linolenic acid promoted BLFX-mediated killing in a dose-dependent manner (*Figure 4C*). Increasing exogenous palmitic acid and linolenic acid increased intracellular palmitic acid and linolenic acid levels,

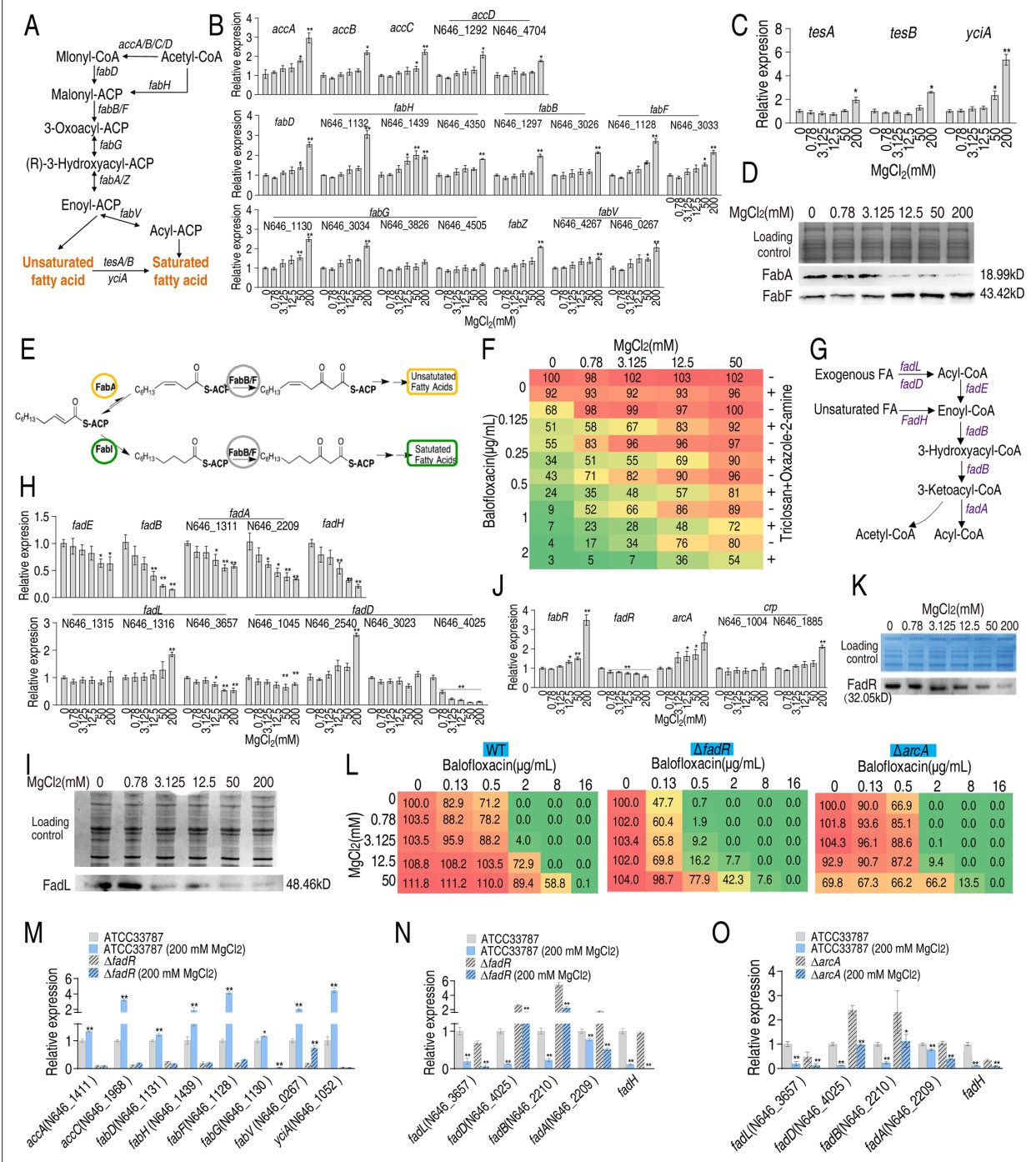

**Figure 3.** Mg²⁺ promotes fatty acid biosynthesis. (**A**) Diagram of fatty acid biosynthesis. (**B**) qRT-PCR for expression of fatty acid biosynthesis genes in the absence or presence of indicated concentrations of MgCl₂ (n = 4). (**C**) qRT-PCR for expression of genes involved in converting unsaturated fatty acids to saturated fatty acids in the absence or presence of indicated MgCl₂ levels (n = 4). (**D**) Western blot for the abundance of proteins responsible for converting unsaturated fatty acids to saturated fatty acids in the absence or presence of indicated concentrations of MgCl₂. (**E**) Diagram of saturated and unsaturated acid biosynthesis. (**F**) Synergy analysis of balofloxacin with triclosan + oxazole-2-amine for ATCC33787 (n = 3). The percentage of bacterial survival was quantified in the presence of indicated MgCl₂ levels and/or balofloxacin plus triclosan and oxazole-2-amine and used to construct the isobologram. Synergy is represented using a color scale or an isobologram, which compares the dose needed for 50% inhibition of synergistic agents (blue) and non-synergistic (i.e., additive) agents (red). (**G**) Diagram of fatty acid degradation. (**H**) qRT-PCR for the expression of genes encoding fatty acid degradation in the absence or presence of the indicated concentrations of MgCl₂ (n = 4). (**I**) Western blot for the abundance of FadL in the absence or presence of the indicated concentrations of MgCl₂. (**J**) qRT-PCR for the expression of fatty acid biosynthesis regulatory genes in the absence or presence of the indicated concentrations of MgCl₂ (n = 4). (**K**) Western blot for the abundance of FadR in the absence or presence of the indicated

*Figure 3 continued on next page*

*Figure 3 continued*

concentrations of MgCl$_2$. (**L**) Synergy analysis for MgCl$_2$ with BLFX for ATCC33787 (WT), Δ*fadR*, and Δ*arcA*. Synergy is represented using a color scale or an isobologram, which compares the dose needed for 50% inhibition of the synergistic agents (blue) and non-synergistic (i.e., additive) agents (red) (n = 3). (**M–NO**) qRT-PCR for expression of genes involved in fatty acid biosynthesis (**M**) and degradation (**N, O**) in ATCC33787, Δ*fadR* or/and Δ*arcA* (**O**) in the presence or absence of 200 mM MgCl$_2$. Whole-cell lysates resolved by SDS-PAGE gel were stained with Coomassie brilliant blue as loading control (**D**), (**I**), and (**K**). Results are displayed as means ± SD, and statistically significant differences are identified by Kruskal–Wallis followed by Dunn's multiple comparison post hoc test unless otherwise indicated. *$p<0.05$ and **$p<0.01$.

The online version of this article includes the following source data for figure 3:

**Source data 1.** File containing original western blots in *Figure 3*, indicating the relevant bands and treatments.

**Source data 2.** Original files for western blot analysis displayed in *Figure 3*.

respectively (*Figure 4D*). When cells were co-treated with palmitic acid, linolenic acid, and BLFX, the two fatty acids appeared to antagonize each other, as demonstrated by the Bliss model (*Figure 4E and F*). Furthermore, magnesium had a minimal effect on the antagonistic effect of palmitic acid, linolenic acid, and BLFX (*Figure 4G*), suggesting that this mineral functions through lipid metabolism. These results indicate that saturated and unsaturated fatty acids have an antagonistic effect on BLFX resistance.

## Mg$^{2+}$ promotes phospholipid biosynthesis

Fatty acids are biosynthetic precursors of lysophosphatidic acid and phosphatidic acid, two key cell membrane components (*Zhang and Rock, 2008a*). Thus, LC-MS was used to assess the effect of Mg$^{2+}$ on membrane lipid composition (*Figure 5—figure supplement 1A*). Higher Mg$^{2+}$ levels increased the percentage of lipids from 61 to 67%, and saturated fatty acids from 24 to 26%, but decreased the percentage of unsaturated fatty acids from 15 to 8% (*Figure 5—figure supplement 1B*). The abundance of 11, 32, and 53 lipids was increased in 3.125, 50, and 200 mM MgCl$_2$-treated bacteria, respectively, while the abundance of 26, 52, and 107 lipids was decreased in 3.125, 50, and 200 mM MgCl$_2$-treated bacteria, respectively (*Figure 5—figure supplement 1C*). Saturated fatty acids and lipids increased and unsaturated fatty acids decreased in an Mg$^{2+}$ dose-dependent manner (*Figure 5—figure supplement 1D*). A total of 52 lipids were quantified, including 17 high-abundance lipids (*Figure 5—figure supplement 1E*). Phosphatidylethanolamine (PE) and phosphatidylglycerol (PG) had the first and second highest abundance (*Figure 5—figure supplement 1E*) and PE levels declined while PG levels increased with increasing Mg$^{2+}$ (*Figure 5A and B*). A similar effect was observed for the unsaturated derivatives of PE and PG (*Figure 5C*). Principal component analysis showed that component t[1] differentiated 0 and 3.125 mM Mg$^{2+}$ from 50 and 200 mM Mg$^{2+}$, while component t[2] separated 50 mM Mg$^{2+}$ from 0 and 200 mM Mg$^{2+}$ and variants in 3.125 mM of Mg$^{2+}$ treatment (*Figure 5D*). S-plot analysis showed that behenic acid, PG[16:1(9Z)/16:1(9Z)], PG[18:2 (9Z, 12Z)/16:0], PA [14:0/22:6 (4Z, 7Z, 10Z, 13Z, 16Z, 19Z)], and lysoPE(16:0/0:0) were upregulated while linoelaidic acid, palmitoleic acid, PE-NMe(24:0/24:0), PE[16:0/14:1(9Z)], and lysoPA(i-12:0/0:0) were downregulated (*Figure 5E*). Of these, the abundance of PG[16:1(9Z)/16:1(9Z)], PG[18:2(9Z,12Z)/16:0], and behenic acid increased, while the levels of linoelaidic acid, palmitoleic acid, PE[16:0/14:1(9Z)], and lysoPA decreased with increasing Mg$^{2+}$ levels (*Figure 5F*). Thus, altered phospholipid abundance may play a role in the effect of Mg$^{2+}$ on antibiotic resistance.

PE and PG are the two end products of phospholipid metabolism (*Figure 5G*). While expression of most genes in the phospholipid biosynthetic pathway, including *psd, pldB,* and *glpQ*, increased in the presence of 200 mM MgCl$_2$, the expression of *pssA* and *etuB, etuC*, encoding the first enzyme and the last enzymes in the pathway, respectively, remained the same. In addition, the expression of *gpsA*, which encodes GpsA and transforms sn-glycerol-3P to glycerone-P, increased independent of Mg$^{2+}$ concentration (*Figure 5H*). Phosphatidylglycerol phosphate synthase (PGS), encoded by *pgsA*, and phosphatidylserine synthase (PSS), encoded by *pssA*, are critical enzymes for the biosynthesis of PG and PE, respectively. PGS and PSS levels were elevated and reduced, respectively, with increasing Mg$^{2+}$ (*Figure 5I*). The effect of Mg$^{2+}$ on the activity of PSS was confirmed using recombinant PSS (*Figure 5J*). Deletion of *plsB* and *pgpA*, the first and last genes involved in PG biosynthesis, respectively, lowered cell viability in the presence of BLFX (note: Δ*psd* could not be obtained) (*Figure 5K and L*). These results indicate that phospholipids may play a role in the effect of Mg$^{2+}$ on BLFX resistance; more specifically, PE may inhibit and PG may promote resistance to BLFX.

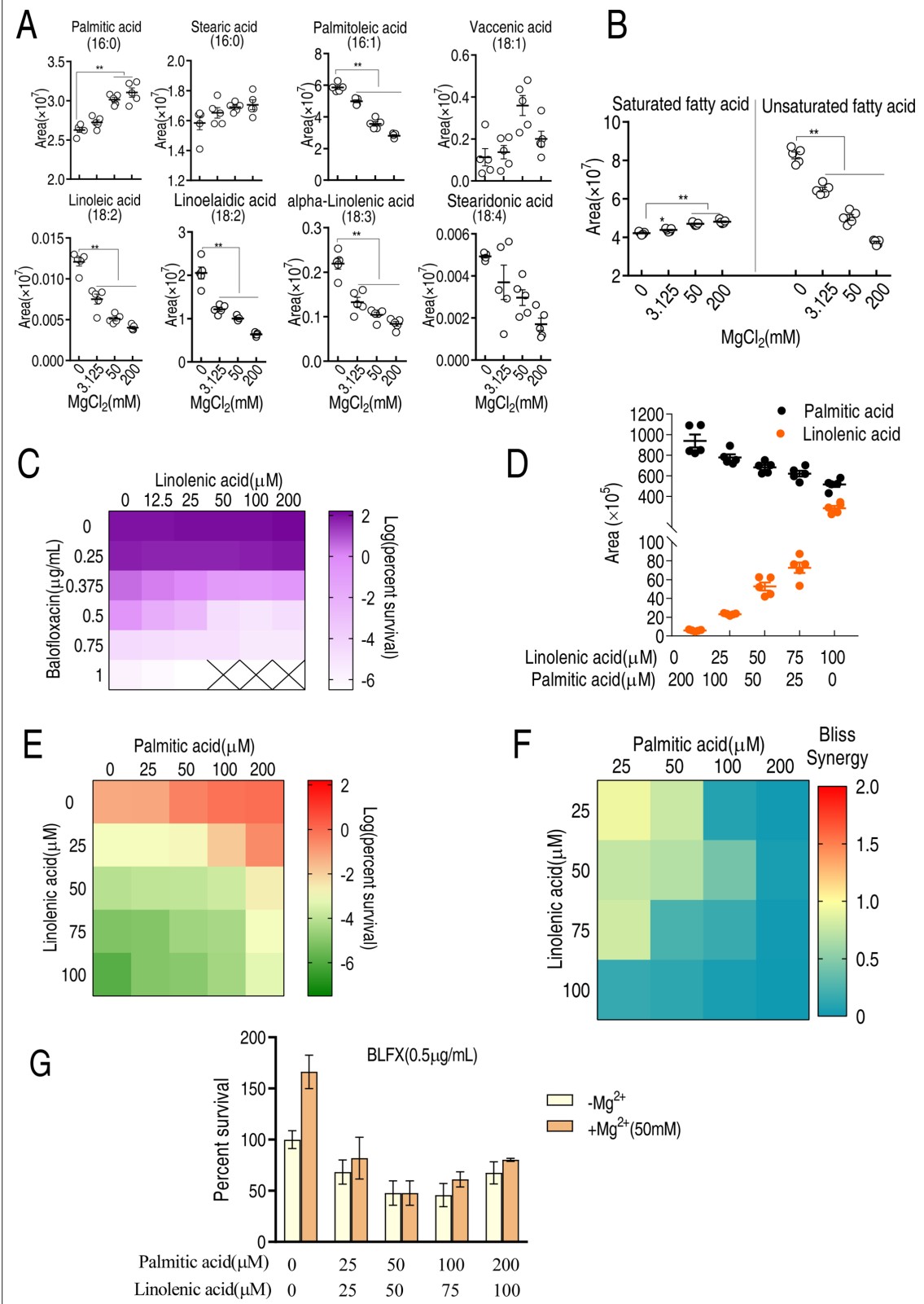

**Figure 4.** LC-MS targeted 16-carbon and 18-carbon fatty acids and the role of palmitic acids and linolenic acids in balofloxacin (BLFX) resistance. (**A**) Scatter plots of 16-carbon and 18-carbon fatty acids, detected by LC-MS. The area indicates the area of the peak of the metabolite in total ion chromatography using GC-MS. (**B**) Scatter plots of total saturated fatty acids and unsaturated fatty acids with 16-carbon and 18-carbon (**A**). (**C**) Synergy analysis for BLFX with linolenic acid for ATCC 33787. Synergy is represented using a color scale or an isobologram, which compares the dose needed

*Figure 4 continued on next page*

*Figure 4 continued*

for 50% inhibition of synergistic agents (while) and non-synergistic (i.e., additive) agents (purple) (n = 3). (**D**) LC-MS for the abundance of intracellular linolenic acid and palmitic acid of ATCC33787 in synergy with the indicated exogenous linolenic acid and palmitic acid. (**E**) Synergy analysis of linolenic acid and palmitic acid in BLFX-mediated killing to ATCC33787. Synergy is represented using a color scale or an isobologram, which compares the dose needed for 50% inhibition of synergistic agents (blue) and non-synergistic (i.e., additive) agents (red). (**F**) Bliss analysis (**E**) (n = 3). (**G**) Percent survival of ATCC33787 in the presence of linolenic acid, palmitic acid, and BLFX with or without 50 mM MgCl$_2$ (n = 3). Results are displayed as means ± SD, and statistically significant differences are identified by Kruskal–Wallis followed by Dunn's multiple comparison post hoc test unless otherwise indicated. *p<0.05 and **p<0.01.

## Mg$^{2+}$ regulates membrane polarization, permeability, and fluidity to confer balofloxacin resistance

Mg$^{2+}$ remodels membrane composition, which may impair membrane potential and polarization, critical to membrane permeability and uptake of antibiotics (*Lee et al., 2019*; *Peng et al., 2015*). The voltage-sensitive dye, DiBAC4(3) showed that 12.5–200 mM MgCl$_2$ promoted membrane depolarization in a dose-dependent manner (*Figure 6A*). Meanwhile, MgCl$_2$ had a dose-dependent (*Figure 6B*) and time-dependent (*Figure 6C*) effect on proton motive force (PMF). These results suggest that Mg$^{2+}$-dependent changes in membrane depolarization may influence antibiotic resistance.

The efficacy of antibiotics is strongly influenced by bacterial membrane permeability and fluidity (*Saeloh et al., 2018*; *Zhao et al., 2021*). Thus, fluorescence microscopy was used to visualize these functions after incubating *V. alginolyticus* cells with FM5-95 and various concentrations of MgCl$_2$. FM5-95 staining decreased with increasing concentrations of Mg$^{2+}$, and no staining was observed in the presence of 200 mM Mg$^{2+}$ (*Figure 6D*). SYTO9, a green fluorescent dye that binds to nucleic acid, enters and stains bacteria cells when there is an increase in membrane permeability (*Lehtinen et al., 2004*; *McGoverin et al., 2020*). Staining decreased with increasing MgCl$_2$, indicating that bacterial membrane permeability declined in an Mg$^{2+}$ dose-dependent manner (*Figure 6E*). These results indicate that exogenous Mg$^{2+}$ reduces bacterial membrane permeability.

Exogenous palmitic acid also shifted the fluorescence signal peaks to the left in an MgCl$_2$-dependent manner while palmitic acid only slightly shifted the peaks (*Figure 6F*). In contrast, exogenous linolenic acid shifted the peak to the right in a dose-dependent manner at 50 mM MgCl$_2$ (*Figure 6G*). These results indicate that exogenous palmitic acid and linolenic acid decrease or increase membrane permeability, respectively. Membrane permeability was increased in Δ*fadR* cells than in WT cells and was reduced by Mg$^{2+}$, remaining higher than the control (*Figure 6H*). However, Δ*arcA* cells had lower membrane permeability at lower MgCl$_2$ concentrations (*Figure 6I*). Δ*pgpA* cells exhibited higher membrane permeability than control cells at all MgCl$_2$ concentrations (*Figure 6J*). These data are consistent with the viability of the mutants described above (*Figures 3L and 5L*).

Relative membrane permeability appeared to correlate with relative intracellular BLFX and antibiotic efficacy. Exogenous palmitic acid and linolenic acid reduced and promoted the uptake of BLFX at 159- and 18-fold, respectively (*Figure 6K*). Loss of *fadR* and *pgpA* increased intracellular BLFX (*Figure 6K*). These data suggest that bacterial membrane permeability is a critical factor required for the efficacy and uptake of BLFX in the presence or absence of exogenous MgCl$_2$.

## Discussion

This study explored the effect of Mg$^{2+}$ on the phenotypic antibiotic resistance of *V. alginolyticus*. Exogenous Mg$^{2+}$ was found to promote saturated fatty acid biosynthesis and inhibit unsaturated fatty acid biosynthesis. Mg$^{2+}$ was also shown to upregulate PGS activity and PG abundance while downregulating PSS activity and PE abundance, decreasing membrane permeability and antibiotic uptake (*Figure 6L*) and enabling phenotypic resistance. These findings are consistent with the known impact of exogenous Mg$^{2+}$ on bacterial survival in the presence of BLFX and other antibiotics, and the association between membrane permeability and drug efficacy. This study further showed that exogenous fatty acids influence membrane permeability and bacterial survival in the presence of BLFX.

Mg$^{2+}$ is the most abundant divalent cation in cells (*Pohland and Schneider, 2019*; *Pontes et al., 2015*) playing many essential roles, including stabilizing macromolecular complexes and membranes, binding cytoplasmic nucleic acids and nucleotides, interacting with phospholipid head groups and cell surface molecules, and acting as an essential cofactor in many enzymatic reactions (*Groisman*

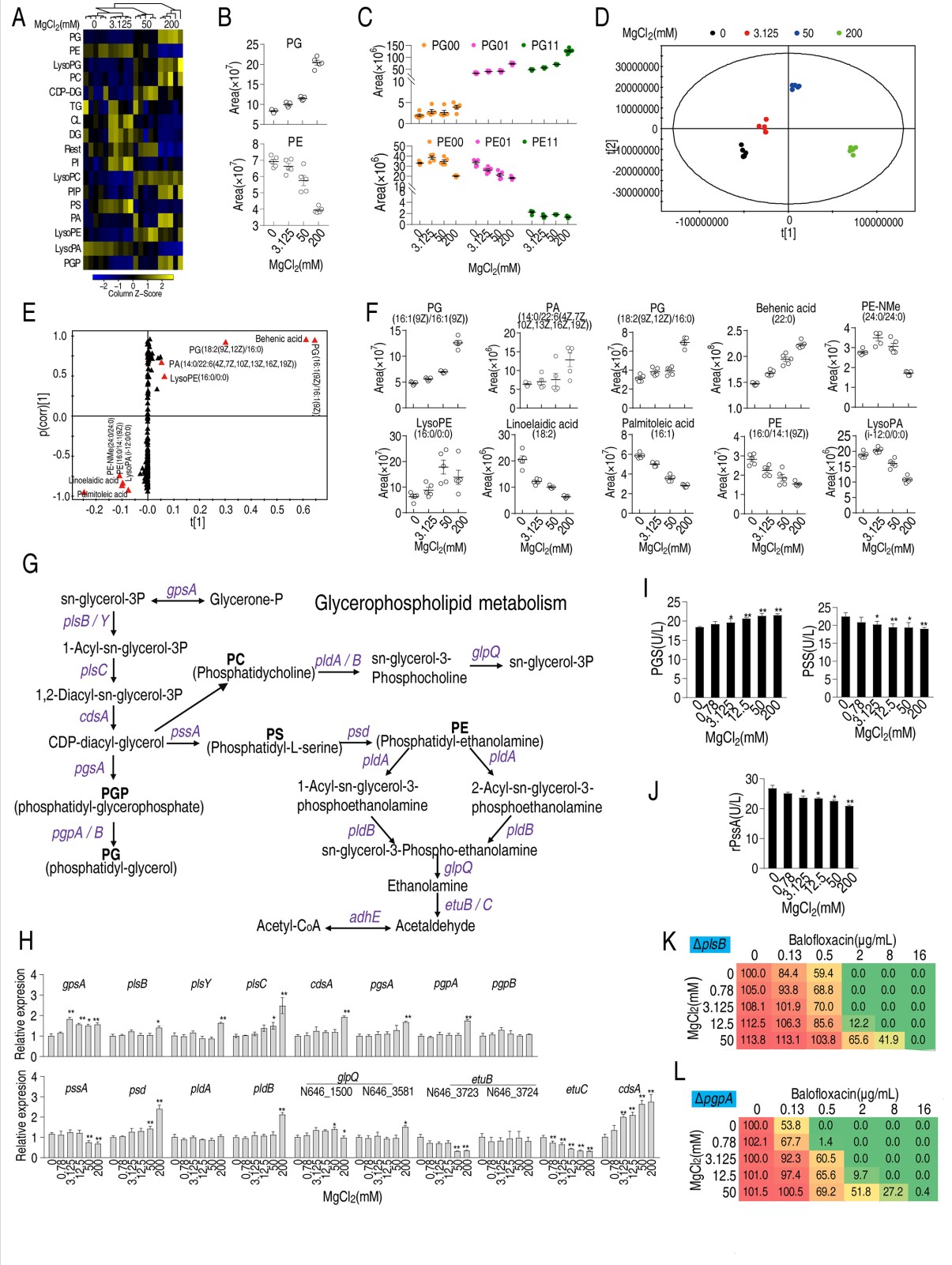

**Figure 5.** Effect of Mg[2+] on phospholipid biosynthesis. (**A**) Heatmap showing changes in differential lipid levels at the indicated concentration of MgCl₂. (**B**) Abundance of ATCC33787 phosphatidylglycerol (PG) and phosphatidylethanolamine (PE) at the indicated concentrations of MgCl₂. (**C**) Scatter plots of PG and PE at different saturation levels in the presence of the indicated MgCl₂ levels. (**D**) Principal component analysis of different concentrations of MgCl₂-induced phospholipids metabolomes. Each dot represents a technical replicate of samples in the plot. (**E**) S-plot generated from orthogonal

*Figure 5 continued on next page*

*Figure 5 continued*

partial least square discriminant analysis (OPLS-DA). Predictive component p[1] and correlation p (corr)[1] differentiated 0 and 3.125 mM $MgCl_2$ from 50 and 200 mM $MgCl_2$. (**F**) Scatter plot of biomarkers in data (**E**). (**G**) Diagram showing glycerophospholipid metabolism. (**H**) qRT-PCR of the expression of genes encoding glycerophospholipid metabolism in the absence of or at the indicated concentrations of $MgCl_2$ (n = 4). (**I**) Phosphatidylglycerol phosphate synthase (PGS) and phosphatidylserine synthase (PSS) levels in the absence or presence of the indicated concentrations of $MgCl_2$ (n = 4). (**J**) Activity of recombinant PSS in the absence or presence of the indicated concentrations of $MgCl_2$ (n = 3). (**K, L**) Synergy analysis for $MgCl_2$ with balofloxacin (BLFX) for Δ*plsB* (**K**) and Δ*pgpA* (**L**) (n = 3). Synergy is represented using a color scale or an isobologram, which compares the dose needed for 50% inhibition for synergistic agents (blue) and non-synergistic (i.e., additive) agents (red). Results are displayed as means ± SD, and statistically significant differences are identified by Kruskal–Wallis followed by Dunn's multiple comparison post hoc test unless otherwise indicated. *p<0.05 and **p<0.01.

The online version of this article includes the following figure supplement(s) for figure 5:

**Figure supplement 1.** Lipidomes in the different concentrations of $MgCl_2$.

*et al., 2013*). The present study showed for the first time that exogenous $Mg^{2+}$ influences palmitic acid and linolenic acid levels and upregulates PGS while downregulating PSS. These findings extend our current understanding of the biological functions of $Mg^{2+}$.

Recent findings suggest a link between fatty acid biosynthesis and the antibiotic resistance of *Edwardsiella tarda* and *V. alginolyticus* (*Su et al., 2021*). The current study found that $Mg^{2+}$ had opposing effects on the abundance of saturated and unsaturated fatty acids, stimulating saturated fatty acid biosynthesis at moderately high levels, while inhibiting unsaturated fatty acids biosynthesis at ≥3 mM. Thus, the ratio of saturated fatty acids to unsaturated fatty acids should help in predicting antibiotic resistance.

Prior studies of Gram-positive and Gram-negative bacteria *Kumariya et al., 2015*; *Said et al., 1987* found that 10–20 mM $Mg^{2+}$ disrupts *Staphylococcus aureus* membranes and kills stationary-phase *S. aureus* cells but does not influence the survival of *E. coli* and *B. subtilis* (*Xie and Yang, 2016*). Low concentrations of $Mg^{2+}$ (≤10 mM) induce PmrAB-dependent modification of lipid A in wild-type *E. coli* (*Herrera et al., 2010*). In two mundticin KS-resistant *Enterococcus faecium* mutants, a putative zwitterionic amino-containing phospholipid increased significantly, while PG and cardiolipin levels declined (*Sakayori et al., 2003*). However, the impact of $Mg^{2+}$ on antibiotic resistance by regulating PGS and PSS activity was not reported. More importantly, exogenous $MgCl_2$ was shown to target these enzymes in opposing ways to increase the PE/PG ratio, a previously unknown mechanism of $Mg^{2+}$-mediated antibiotic resistance.

Prior studies show that in *Stenotrophomonas maltophilia,* PhoP/Q is activated by low magnesium levels and PhoP/Q inhibition or inactivation stimulates β-lactam antibiotic uptake (*Huang et al., 2021*). The effects of the outer membrane permeabilizers, polymyxin B nonapeptide and EDTA, are completely abolished by 3 mM $Mg^{2+}$ (*Kwon and Lu, 2006*). In response to $Mg^{2+}$-limited growth, enteric Gram-negative bacteria show higher lipid A acylation, which alters membrane permeability and reduces the uptake of cationic antimicrobial peptides (*Guo et al., 1998*). $Mg^{2+}$ reduces the high sensitivity of rough *Salmonella typhimurium, S. minnesota,* and *E. coli* 08 (which includes defects in the lipopolysaccharide carbohydrate core) mutants to several antibiotics (*Stan-Lotter et al., 1979*). $Mg^{2+}$ (1 mM) inhibited aminoglycoside-mediated outer membrane permeabilization in *P. aeruginosa* (*Hancock et al., 1981*). However, mechanisms underlying the impact of $Mg^{2+}$ on antibiotic resistance have remained largely unknown. The current study revealed that $Mg^{2+}$ plays a critical role in regulating the membrane permeability required for antibiotic resistance by modulating PE and PG during phospholipid metabolism.

Aquaculture is an environmental gateway to the development and globalization of antimicrobial resistance due to the excessive use of these drugs to prevent and treat bacterial contaminants (*Cabello et al., 2020*). The current study found that $Mg^{2+}$ increased antibiotic resistance, providing a reasonable explanation of why antibiotics are being excessively used in aquaculture and a possible solution to $Mg^{2+}$-induced antibiotic resistance by balancing phospholipid metabolism. This finding should promote an overall reduction in antibacterial use that will improve environment and food safety.

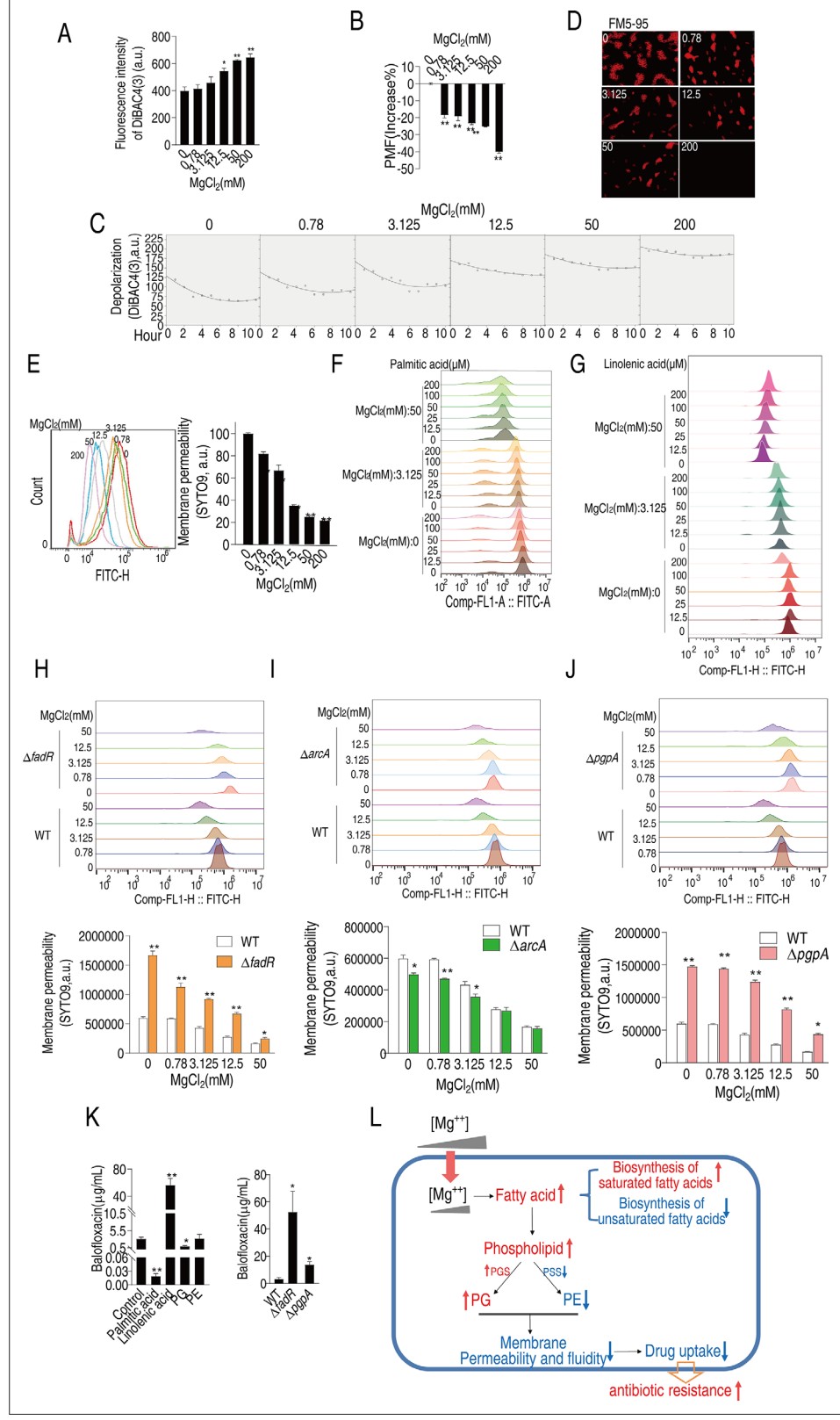

**Figure 6.** Mg$^{2+}$ regulates membrane polarization, permeability, and fluidity to confer balofloxacin (BLFX) resistance. (**A, B**) Depolarization (**A**) and proton motive force (PMF) (**B**) of ATCC33787 in the absence of or at the indicated concentrations of MgCl$_2$ (n = 3). (**C, D**) Dynamic depolarization. (**D**) Membrane fluidity of ATCC33787 in the absence or presence of indicated concentrations of MgCl$_2$, as shown by fluorescence microscopy.

*Figure 6 continued on next page*

*Figure 6 continued*

(**E**) Membrane permeability of ATCC33787 in the absence or presence of the indicated concentrations of $MgCl_2$ (n = 3). (**F, G**) Membrane permeability of ATCC33787 cultured in palmitic acid (**F**) or linolenic acid (**G**) at the indicated concentrations of $MgCl_2$ (n = 3). (**H, I**) Membrane permeability of Δ*fadR* (**H**) and Δ*arcA* (**I**) in the absence or presence of the indicated concentrations of $MgCl_2$ (n = 3). (**J**) Membrane permeability of Δ*pgpA* in the absence or presence of the indicated concentrations of $MgCl_2$ (n = 3). (**K**) Intracellular BLFX of ATCC33787 in the presence of palmitic acid, linolenic acid, phosphatidylglycerol (PG), and phosphatidylethanolamine (PE) (left panel) or Δ*fadR* and Δ*pgpA* mutants (right panel) (n = 3). (**L**) Diagram of mechanisms of $Mg^{2+}$-mediated resistance to BLFX. Results are displayed as means ± SD, and statistically significant differences are identified by Kruskal–Wallis followed by Dunn's multiple comparison post hoc test unless otherwise indicated. *p<0.05 and **p<0.01.

In summary, $Mg^{2+}$ modulated the phenotypic resistance of *V. alginolyticus* to BLFX and potentially other quinolones and other classes of antibiotics by altering membrane permeability and thus reducing antibiotic uptake. These findings inform the development of methods to improve the therapeutic effect of antibiotics on magnesium-induced phenotypic resistance.

# Materials and methods

## Bacterial strains and culture

*V. alginolyticus* ATCC33787 and *V. parahaemolyticus* 01 are from our laboratory collection (*Jiang et al., 2022*; *Kou et al., 2022*). They were cultured in 0.5% yeast broth (HuanKai Microbial, Guangdong, China) (pH 7.0) with 3% NaCl at 30°C overnight. The cultures were diluted 1:50 (v/v) and grown in fresh 0.5% yeast broth supplemented with desired concentrations of $MgCl_2$ at 30°C until $OD_{600}$ = 0.6. Bacteria were harvested by centrifugation at 8000 rpm for 3 min and resuspended in corresponding concentrations of $MgCl_2$ to 0.6 of $OD_{600}$.

## Measurement of the MIC using the microtiter-dilution method

MIC using the microtiter-dilution method was performed as previously described (*Zhang et al., 2019*). In brief, the overnight bacterial cultures in 3% NaCl were diluted at 1:100 (v/v) in fresh 0.5% yeast broth supplemented with desired concentrations of $MgCl_2$, cultured at 30°C and collected when the cells arrived at an $OD_{600}$ of 0.6 in medium without $MgCl_2$. Then, $1 \times 10^5$ CFU cells were dispensed into each well of a 96-well microtiter polystyrene tray after which a series of twofold dilutions of antibiotic was added. Following incubation at 30°C for 16 hr, the MIC was defined as the lowest antibiotic concentration that inhibited visible growth. Three biological repeats were carried out.

## Measurement of MIC by Oxford method

Determination of MIC by Oxford cup was performed as previously described (*Liu et al., 2015*). Bacterial cells cultured overnight in yeast medium were diluted at 1:100, shaken at 30°C and 200 rpm until the $OD_{600}$ nm reached at 0.6. Aliquot of 100 μL cells were spread on 0.5% yeast solid medium containing 3% NaCl and 0, 0.625, 1.25, 2.5, 5, 10, 20, or 40 mM of $MgCl_2$. Oxford cups were placed on the solid medium, and various amounts of BLFX (0, 0.39, 0.78, 1.56, 3.125, 6.25, and 12.5 μg) were added. After culturing at 30°C for 12 hr, diameter of the inhibition zone was measured and the inhibition area was calculated.

## Measurement of intracellular balofloxacin

Measurement of intracellular BLFX was performed by LC-MS analysis and plate-counting assay. For LC-MS analysis, ATCC33787 cultured in the desired concentrations of $MgCl_2$ were collected and adjusted to OD 0.6. Aliquot of 50 mL bacterial cells were added into a 250 mL Erlenmeyer flask and then 300 μL of 10 mg/mL BLFX (final concentration is 60 μg/mL). After being cultured for 6 hr at 30°C, these bacterial cells were collected and adjusted to OD 1.0 with 3% NaCl containing corresponding concentrations of $MgCl_2$. Aliquots of 30 mL bacteria were collected and washed three times using mobile phase (acetonitrile:double distilled water containing 0.1 mol/l formic acid = 35:65) for LC-MS detection. The bacterial cells were added 1 mL mobile phase, crushed in ice water bath for 10 min (crush for 2S, pause for 3S at 35% power). Following centrifugation, supernatants were collected, filtered, and then measured using liquid chromatography for BLFX. Different concentrations of BLFX

were used for a standard curve. For plate-counting assay, ATCC33787 cells cultured overnight were diluted at 1:100, shaken at 30°C and 200 rpm until the $OD_{600}$ nm reached 0.6. The cells were diluted at 1:100 and spread on LB agar to prepare ATCC33787 plates. Meanwhile, ATCC33787 cells were cultured in medium with desired $MgCl_2$ and at 30°C and 200 rpm for 6 hr. These cells were collected, washed, and sonicated. The sonicated solution was added to Oxford cups with the ATCC33787 plates. After culturing at 30°C for 12 hr, the diameter of the inhibition zone was measured. A gradient of BLFX was used as a standard curve for drug quantification.

## Measurement of intracellular magnesium

Measurements of intracellular $Mg^{2+}$ concentration were carried out as previously described with a modification (*Yang et al., 2018*). In brief, *V. alginolyticus* cultured in 0, 0.78, 3.125, 12.5, 50, and 200% $MgCl_2$ were collected, washed, and resuspended in the same concentrations of $MgCl_2$ to an $OD_{600}$ of 1.0. Aliquots of 20 mL of bacterial suspensions were centrifuged and washed by $ddH_2O$ once. The resulting cells were weighted, which was designated wet weight. These samples were freeze-dried overnight, which was designated dry weight. Intracellular water volume was calculated using the following formula: W × (1W/D-0.23) as described previously (*Unemoto et al., 1973*). 200 µL of concentrated nitric acid was added to the dried cells and then heated at 75°C for 20 min. After diluting the samples 20-fold, they were analyzed for $Mg^{2+}$ concentration by inductively coupled plasma mass spectrometry (iCAP 6500, Thermo Fisher). The intracellular $Mg^{2+}$ concentration was calculated according to the volume of intracellular water.

## Western blotting

Western blotting was carried out as described previously (*Jiang et al., 2023a*; *Yao et al., 2019*). Bacterial protein samples were prepared by ultrasound treatment, resolved on a 12% SDS-PAGE, and transferred to nitrocellulose membranes (GE Healthcare Life Sciences). The membranes were incubated with 1:100 of the primary mouse antibodies, followed by goat anti-mouse secondary antibodies conjugated with horseradish peroxidase. Band intensities were detected using a chemiluminescence imaging analysis system, Tanon-5200.

## Metabolomics analysis

Metabolomics analysis was performed by GC-MS as described previously (*Jiang et al., 2023b*; *Yang et al., 2018*). Briefly, ATCC33787 were cultured in 0.5% yeast broth with desired $MgCl_2$. Equivalent numbers of cells were quenched with 60% (v/v) cold methanol (Sigma) and then centrifuged at 8000 rpm at 4°C for 5 min. 1 mL of cold methanol was used to extract metabolites. To do this, the samples were sonicated for 5 min at a 10-Wpower setting using the Ultrasonic Processor (JY92-IIDN, Scientz, China), followed by centrifugation at 12,000 rpm at 4°C for 10 min. Supernatants were collected and 10 µL ribitol (0.1 mg per mL, Sigma-Aldrich, USA) was added into each sample as an internal quantitative standard. The supernatants were concentrated for metabolite derivatization and then used for GC-MS analysis. Every experiment was repeated by five biological replicates. GC-MS detection and spectral processing for GC-MS were carried out using the Agilent 7890A GC equipped with an Agilent 5975C VL MSD detector (Agilent Technologies, USA) as described previously (*Unemoto et al., 1973*; *Yang et al., 2018*). Statistical difference was obtained by Kruskal–Wallis test and Mann–Whitney test using SPSS 13.0 and a p-value 0.01 was considered significant. Hierarchical clustering was completed in the R platform (https://cran.r-project.org/) with the function 'heatmap. 2' of 'gplots library'. Z score analysis was used to scale each metabolite. Multivariate statistical analysis included OPLS-DA implemented with SIMCA 12.0 (Umetrics, Umeå, Sweden). Control scaling was selected prior to fitting. All variables were mean centered and scaled to pareto variance of each variable. Rank-sum test and a permutation test were used to identify differentially expressed metabolites. OPLS-DA was used to reduce the high dimension of the data set. Differential metabolites to their respective biochemical pathways were outlined in the MetaboAnalyst 3.0 (http://www.metaboanalyst.ca/). Pathways were enriched by raw p-value <0.05.

## qRT-PCR

Quantitative real-time PCR (qRT-PCR) was carried out as described previously (*Yang et al., 2020*). Total RNA was extracted from *V. alginolyticus* using TRIZOL regent (Invitrogen Life Technologies)

according to the protocol. RNA electrophoresis was carried out in 1% (w/v) agarose gels to identify quality of the extracted RNA. By using a PrimeScript RT reagent Kit with gDNA eraser (Takara, Japan), reverse transcription-PCR was carried out on 1 µg of total RNA and primers are listed in *Supplementary file 1b*. qRT-PCR was performed in 384-well plates with a total volume of 10 µL, and the reaction mixtures were run on a LightCycler 480 system (Roche, Germany). Data are shown as the relative mRNA expression compared to 0% $MgCl_2$ test with the endogenous reference 16S rRNA gene.

## Antibiotic bactericidal assay

Antibiotic bactericidal assay was performed as described previously with a modification (*Li et al., 2016*; *Zhao et al., 2021*). The cultured bacteria of ATCC33787 and its mutant strains were transferred to fresh yeast medium at a dilution of 1:1000 and dispensed into test tubes and then the indicated concentrations of $MgCl_2$ and BLFX were added. If desired, 2 mM 2-aminooxazole and 1 µg/mL triclosan were used or a metabolite complemented. These mixtures were incubated at 30°C and 200 rpm for 6 hr. Cells were collected. To determine CFU per mL, 100 µL samples were tenfold serially diluted with 900 µL M9 buffer and an aliquot of 5 µL of each dilution was spotted onto the LB agar plates and cultured at 30°C for 8 hr.

## Construction of gene-deleted mutants

Construction of gene-deleted mutants was carried out as described previously (*Kuang et al., 2021*). Primers were designed as shown in *Supplementary file 1c* using CE Design V1.03 software. To construct gene-deleted mutants, upstream and downstream 500 bp fragments were amplified from the genome using two pairs of primers (primers P1 and P2, primers P3 and P4), and then merged into a 1000 bp fragment by overlap PCR using a pair of primers (primers P1 and P4). After the fragments were digested by *Xba*I, they were ligated into the pDS132 vector digested by the same enzymes and transformed into MC1061 competent cells. Conversion products were coated on LB plate containing 25 µg/mL chloramphenicol and cultured at 37°C overnight. The plasmids from colony growing on the plate were identified by PCR using a pair of primers (primers P1 and P4) and sequenced. The sequenced plasmids were transformed into MFD $\lambda$ pir competent cells as donor. MFD $\lambda$ pir and recipient bacterium ATCC 33787 were cultured to an optical density (OD) of 1.0 and then mixed at a ratio of 4:1. After centrifugation, the pellets were resuspended with 50 µL LB medium including DAP (100 µg/mL), dropped onto sterilized filter paper on LB medium including DAP (100 µg/mL), and cultured for 16–18 hr at 37°C.

All bacteria rinsed from the filter paper with LB medium were smeared onto the LB plate with chloramphenicol (25 µg/mL) and ampicillin (100 µg/mL). After cultured for 16–18 hr at 37°C, bacteria were screened by LB plate with the above two antibiotics. The bacteria were identified by plasmid PCR using a pair of primers P1 and P4 and sequencing and then were cultured and smeared onto the LB plates with 20% sucrose. The clones were cultured and smeared onto the LB plates with 20% sucrose again. The clones that did not grow on the LB plates with chloramphenicol but grew on the LB plates with 20% sucrose were identified by PCR using primer P7P8 (primer P7 is set at about 250 bp upstream of P1, and primer P8 is set at about 250 bp downstream of P4), P4P7, and P5P6 (for amplification of the target gene).

## LC-MS analysis for lipidomics

ATCC33787 were cultured in medium with desired concentrations of $MgCl_2$, washed by 3% NaCl with the desired concentrations of $MgCl_2$, and adjusted to OD 1.0. Aliquot of 30 mL, the cultures was centrifuged each sample and bacterial cells were collected. Aliquot of 800 µL distilled water was added and treated for 5 min at 100°C to inactivate phospholipase C. Protein concentration was determined with BCA kit. Sample solution with 3 mg protein was transferred into a 10 mL centrifuge tube and supplemented with distilled water to 800 µL. Then all samples were processed as follows: 1 mL chloroform and 2 mL methanol (to make the volume ratio of chloroform:methanol:water = 1:2:0.8) were added and cholic acid was used as an internal standard. These mixtures were vortexed for 2 min and then 1 mL chloroform was added and vortexed for 30 s; 1 mL 10% NaCl solution was added, vortexed for 30 s, and placed at room temperature overnight. After the solution was layered, a 2.5 mL syringe was used to suck the lowest layer (chloroform layer) to a new EP tube. Solution of the chloroform layer was evaporated in the rotary evaporator for 2 hr and then 1 mL mobile phase (50% and

50% B. A, 25 mM ammonium acetate/methanol [30:70]; B, methanol) was added for analysis of lipids by LC-MS. Database (https://hmdb.ca) was used for lipid identification.

## Measurement of enzyme activity

Activity of pyruvate dehydrogenase (PDH), α-ketoglutarate dehydrogenase (KGDH), and succinate dehydrogenase (SDH) was carried out as previously described (*Jiang et al., 2020*). Cells cultured in medium were collected and washed three times with saline. The bacterial cells were suspended in Tris–HCl (pH 7.4) and disrupted by sonic oscillation for 6 min (200 W total power with 35% output, 2 s pulse, 3 s pause over ice). After centrifugation, supernatants were collected. The protein concentration of the supernatant was determined using Bradford assay (Beyotime, P0009). Then, 200 μg proteins were used for the determination of PDH, KGDH, and SDH activity. Levels of PGS and PSS were quantified by ELISA kits according to the manufacturer's instruction (Shanghai Fusheng Industrial Co., Ltd., China).

## Measurement of proton motive force

Measurement of the transmembrane voltage delta PSI (δPSI), which is the electrical component of the PMF, was performed as previously described (*Cheng et al., 2018*). Bacteria cultured in medium with desired concentrations of $MgCl_2$ were collected at centrifugation and labeled by DiO2(3). Approximately $1 \times 10^7$ CFU were added to a flow cytometry tube containing 1 mL buffer with 10 μM DiOC2(3) (Sigma) and incubated in the dark for 30 min at 30°C. Samples were assayed with BD FACSCalibur flow cytometer with a 488 nm excitation wavelength. Gates for bacterial populations were based on the control population by using forward versus side scatter and red versus green emission. Size and membrane potential determined the intensity of red (488 nm excitation, 610 nm emission) fluorescence. The diverse ratios of red and green indicated fluorescence intensity values of the gated populations. Computational formula of membrane potential: $\text{Log}\left(10^{3/2} \times \left(\frac{\text{red fluorescence}}{\text{green fluorescence}}\right)\right)$.

## Measurement of depolarization

The membrane potential was estimated by measuring the fluorescence of the potential-dependent probe DiBAC4(3) (*Saint-Ruf et al., 2016*). DiBAC4(3) is a s voltage-sensitive probe that penetrates depolarized cells, binding intracellular proteins or membranes exhibiting enhanced fluorescence and red spectral shift. ATCC33787 were cultured in medium with desired concentrations of $MgCl_2$, collected and then adjusted to OD 0.6. Aliquot of 100 μL bacterial cells were diluted to 1 mL, and 2 μL of 5 mM DiBAC4(3) was added. After incubated for 15 min at 30°C without light and vibration, these samples were filtered and detected by BD FACSCalibur.

## Measurement of fluidity by fluorescence microscopy

Measurement of membrane fluidity is performed as previously described (*Wen et al., 2022*). Briefly, ATCC33787 were cultured in medium with indicated concentrations of $MgCl_2$, collected, and then adjusted to OD 0.6. Aliquots of 100 μL bacteria cells of each sample were diluted to 1 mL, and 10 μL (10 mg/mL) FM5-95 (Thermo Fisher Scientific, USA) was added. FM5-95 is a lipophilic styryl dye that is inserted into the outer leaflet of bacterial membrane and becomes fluorescence. This dye preferentially binds to the microdomains with high membrane fluidity (*Wen et al., 2022*). After incubated for 20 min at 30°C at vibration without light, the sample was centrifuged for 10 min at 12,000 rpm. The pellets were resuspended with 20 μL of 3% NaCl. Aliquots of 2 μL sample were dropped on the agarose slide and photos taken under the inverted fluorescence microscope.

## Measurement of membrane permeability

Measurement of membrane permeability was carried out as described previously (*Chen et al., 2025*; *Su et al., 2021*). ATCC33787 were cultured in medium with desired concentrations of $MgCl_2$, collected, and then adjusted to OD 0.6. Aliquots of 100 μL bacteria cells of each sample were diluted to 1 mL

and 2 μL 10 mg/mL SYT09 was added. After incubated for 15 min at 30°C at vibration without light, the mixtures were filtered and measured by flow cytometry (BD FACSCalibur, USA).

## Acknowledgements

This work was sponsored by National Natural Science Foundation of China (32273177) (to PB), Innovation Group Project of Southern Marine Science and Engineering Guangdong Laboratory (Zhuhai) (311020006) (to PB), Natural Science Foundation of Guangdong Province (2022A1515012079) (to LH), the Science and Technology Planning Project of Guangdong Province (2023B1212060028), and Fundamental Research Funds for the Central Universities, Sun Yat-sen University (24lgzy004).

## Additional information

### Funding

| Funder | Grant reference number | Author |
|---|---|---|
| National Natural Science Foundation of China | 32273177 | Bo Peng |
| Innovation group project of Southern Marine Science and Engineering Guangdong Laboratory | 311020006 | Bo Peng |
| Natural Science Foundation of Guangdong Province | 2022A1515012079 | Hui Li |
| The Science and Technology Planning project of Guangdong Province | 2023B1212060028 | Bo Peng |
| Fundamental Research Funds for the Central Universities, Sun Yat-sen University | 24lgzy004 | Bo Peng |

The funders had no role in study design, data collection and interpretation, or the decision to submit the work for publication.

### Author contributions

Hui Li, Data curation, Formal analysis, Funding acquisition, Investigation, Writing – review and editing; Jun Yang, Data curation, Formal analysis, Validation, Investigation, Methodology; Su-fang Kuang, Hui-yin Lin, Validation, Investigation; Huan-zhe Fu, Formal analysis, Investigation; Bo Peng, Conceptualization, Formal analysis, Supervision, Funding acquisition, Writing – original draft, Project administration, Writing – review and editing

### Author ORCIDs

Bo Peng (ID) https://orcid.org/0000-0002-5698-6097

Reviewer #1 (Public review): https://doi.org/10.7554/eLife.100427.3.sa1
Reviewer #2 (Public review): https://doi.org/10.7554/eLife.100427.3.sa2
Author response https://doi.org/10.7554/eLife.100427.3.sa3

## Additional files

### Supplementary files

Source data 1. File containing original western blots in Appendix 1, indicating the relevant bands and treatments.

Source data 2. Original files for western blot analysis displayed in Appendix 1.

Supplementary file 1. Gradients of medium and primers used in this study. (a) Comparison in components in LBS and ASWT. (b) Primes used in the present study. (c) Primers used in the present study for the construction of gene-deleted mutants.

MDAR checklist

## Data availability

Data is available in the manuscript and supporting files. All source data have been deposited to Figshare at: https://doi.org/10.6084/m9.figshare.28014866.

The following dataset was generated:

| Author(s) | Year | Dataset title | Dataset URL | Database and Identifier |
|---|---|---|---|---|
| Li H, Yang J, Kuang S-F, Fu H-Z, Lin H-Y, Peng B | 2024 | Magnesium modulates phospholipid metabolism to promote bacterial phenotypic resistance to antibiotics | https://doi.org/10.6084/m9.figshare.28014866.v2 | figshare, 10.6084/m9.figshare.28014866.v2 |

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

## Appendix 1

### Magnesium and balofloxacin binding only partially contributes to phenotypic resistance

Prior studies suggest that the chelation of antibiotics by magnesium ions inhibits antibiotic uptake (*Deitchman et al., 2018*; *Lunestad and Goksøyr, 1990*). To investigate whether magnesium binds to BLFX, BLFX was preincubated with magnesium, and zone of inhibition (ZOI) analysis was conducted. Six different concentrations of BLFX were separately incubated with six different concentrations of $MgCl_2$, and then spotted on filter paper so that a defined amount of BLFX could be used for ZOI. While lower concentrations of $MgCl_2$ (0.78, 3.125, or 12.5 mM) did not alter the ZOI, higher concentrations, including 50 and 200 mM $MgCl_2$, decreased the ZOI (*Appendix 1—figure 1A*), suggesting that even high doses of magnesium had only a partial effect on BLFX through direct binding. For example, at 200 mM $MgCl_2$ and 5 or 10 µg/mL BLFX, the BLFX ZOI was 53.2 and 70.3% of the ZOI at 0 mM $MgCl_2$, suggesting that ≥50% of the antibiotics were still functional. Intracellular BLFX also decreased with increasing $MgCl_2$ (*Appendix 1—figure 1B*), while exogenous $Mg^{2+}$ increased intracellular $Mg^{2+}$ levels in a dose-dependent manner. For example, exogenous 50 and 200 mM $MgCl_2$ increased intracellular $Mg^{2+}$ levels to 1.21 and 1.31 mM, respectively (*Appendix 1—figure 1C*). The relationship between TolC, an efflux pump that transports quinolones from bacterial cells, and $Mg^{2+}$ was also assessed (*Kobylka et al., 2020*). The expression of TolC/tolC was unaffected by $Mg^{2+}$ (*Appendix 1—figure 1D*). Magnesium is critical for LPS stability. LPS levels increased at 200 mM $Mg^{2+}$ (*Appendix 1—figure 1E*), however, the loss of waaF, lpxA, and lpxC, three key genes involved in LPS biosynthesis, did not influence BLFX sensitivity/resistance in the presence of $Mg^{2+}$ (*Appendix 1—figure 1F*). These findings suggest that magnesium-induced LPS biosynthesis does not contribute directly to BLFX resistance and demonstrate that $Mg^{2+}$ influx is involved in BLFX resistance.

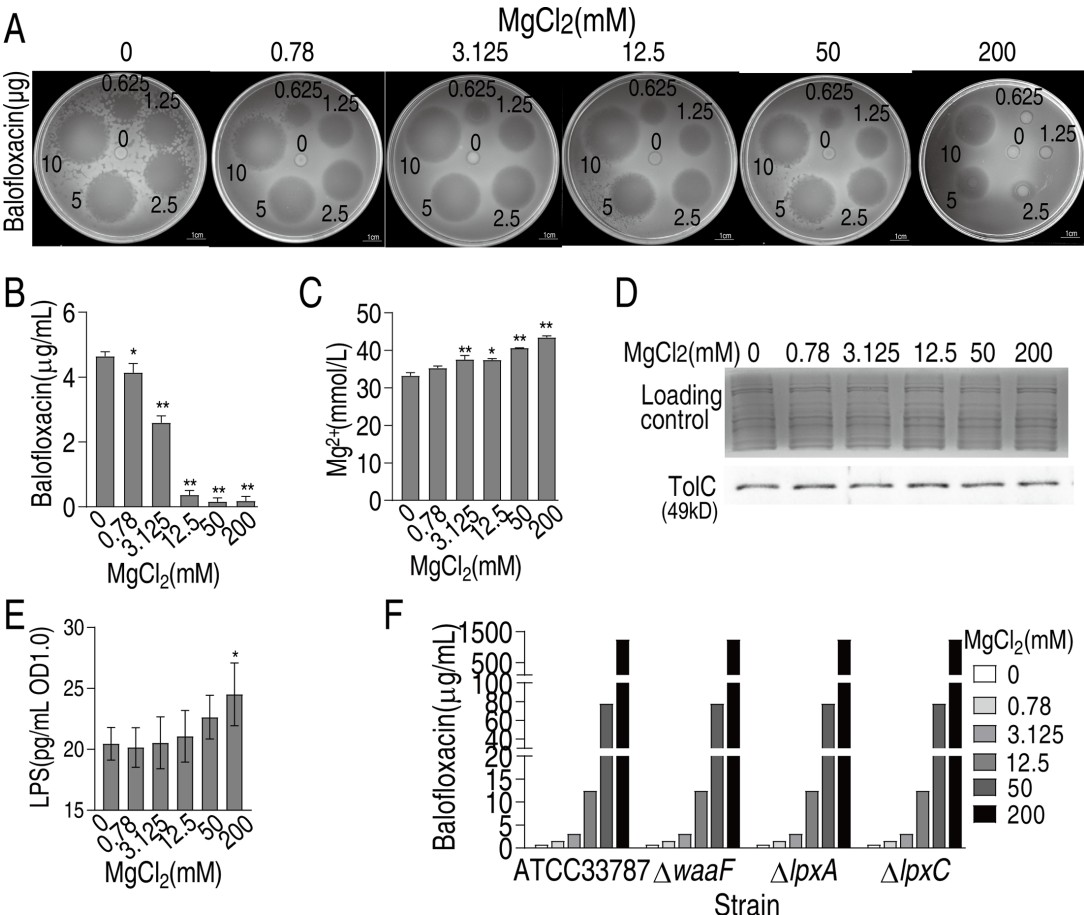

**Appendix 1—figure 1.** $Mg^{2+}$ decreases balofloxacin (BLFX) uptake. (**A**) Oxford cup test for effect of different concentrations of $Mg^{2+}$ on antibacterial action of different dose of BLFX to ATCC33787. To do this, 0, 0.78, 3.125, 12.5, 50, and 200 mM $MgCl_2$ were individually mixed with 0, 12.5, 25, 50, 100, or 200 µg/mL BLFX for 5 hr. Then, 50 µL were added into Oxford cup for antibacterial efficiency, which contained 0, 0.625, 1.25, 2.5, 5, or 10 µg BLFX, respectively. (**B**) Intracellular BLFX of ATCC 33787 in artificial seawater (ASWT) with the indicated concentrations of $MgCl_2$ and 60 µg/mL BLFX. (**C**) Intracellular $Mg^{2+}$ of ATCC 33787 in ASWT with the indicated concentrations of $MgCl_2$. (**D**) Western blot for abundance of TolC in the presence of $MgCl_2$. Whole-cell lysates resolved by SDS-PAGE gel were stained with Coomassie brilliant blue as loading control. (**E**) LPS quantification at the indicated concentrations of $MgCl_2$. (**F**) MIC of ATCC 33787 and its mutants $\Delta waaF$、$\Delta lpxA$、$\Delta lpxC$ in ASWT with the indicated concentrations of $MgCl_2$, which is measured using the microtiter-dilution method.

The online version of this article includes the following source data for appendix 1—figure 1:

**Appendix 1—figure 1—source data 1.** PDF file containing original western blots in 1 *Appendix 1—figure 1*, indicating the relevant bands and treatments.

**Appendix 1—figure 1—source data 2.** Original files for western blot analysis displayed in *Appendix 1—figure 1*.

