## [Editor Report · eLife Assessment]

The study explored the influence of magnesium on phenotypic antibiotic resistance in two strains of *Vibrios*: *V. alginolyticus* ATCC33787 and *V. parahaemolyticus* VP01. This research is **fundamental** for revealing the phenotypic antibiotic resistance mechanism utilized by the specified model bacteria in elevated levels of magnesium. The study produced **convincing** evidence indicating that in high concentrations of magnesium, the efficacy of selected antibiotics was diminished due to decreased biosynthesis of unsaturated fatty acids and PE, along with an increase in the biosynthesis of PG.

---

## [Referee Report · Reviewer #1 (Public review)]

Summary:

In the manuscript entitled "Magnesium modulates phospholipid metabolism to promote bacterial phenotypic resistance to antibiotics", Li et al demonstrated the role of magnesium in promoting phenotypic resistance in V. alginolyticus. Using standard microbiological and metabolomic techniques, the authors have shown the significance of fatty acid biosynthesis pathway behind the resistance mechanism. This study is significant as it sheds light on the role of an exogenous factor in altering membrane composition, polarization and fluidity which ultimately leads to antimicrobial resistance.

Strengths:

Authors have used different approaches to demonstrate the effect of Mg+2 on drug resistance in Vibrio alginolyticus. The revised version of the manuscript is much improved, with a very informative introduction and a variety of methodologies with clear explanation of the experiments performed. Also, additional experiments were performed as suggested by the reviewers which certainly enhanced the quality of the paper. I believe the findings of this study will be of high impact in the bacterial community.

Weaknesses:

There are a few grammatical mistakes.

Comments on revisions:

The authors have done a comprehensive job of addressing all my concerns in their revised version.

---

## [Referee Report · Reviewer #2 (Public review)]

Summary:

In this study, the authors aimed to identify if and how magnesium affects the ability of two particular bacteria species to resist the action of antibiotics. In my view, the authors succeeded in their goals and present a compelling study that will have important implications for the antibiotic resistance research community. Since metals like magnesium are present in all lab media compositions and are present in the host, the data presented in this study certainly will inspire additional research by the community. These could include research into whether other types of metals also induce multi-drug resistance, whether this phenomenon can be observed in other bacterial species, especially pathogenic species that cause clinical disease, and whether the underlying molecular determinants (i.e. enzymes) of metal-induced phenotypic resistance could be new antimicrobial drug targets themselves.

Strengths:

This study's strengths include that the authors used a variety of methodologies, all of which point to a clear effect of exogenous Mg2+ on drug resistance in the targeted species. I also comment the authors for carrying out a comprehensive study, spanning evaluation of whole cell phenotypes, metabolic pathways, genetic manipulation, to enzyme activity level evaluation. The fact that the authors uncovered a molecular mechanism underlying Mg2+-induced phenotypic resistance is particularly important as the key proteins should be studied further.

Weaknesses:

I thank the authors for improving their manuscript based on my previous suggestions. I still believe the Results section is long and bogs down at times.

In general, the conclusions drawn by the authors are justified by the data, except for the interpretation of some experiments. Importantly, this paper has discovered new antimicrobial resistance mechanisms and has also pointed to potential new targets for antimicrobials.

Comments on revisions:

I just wanted to thank the authors for addressing most of my previous comments.

---

## [Author Response]

The following is the authors’ response to the original reviews.

**Public Reviews:**

**Reviewer #1 (Public Review):**
Summary:In the manuscript entitled "Magnesium modulates phospholipid metabolism to promote bacterial phenotypic resistance to antibiotics", Li et al demonstrated the role of magnesium in promoting phenotypic resistance in V. alginolyticus. Using standard microbiological and metabolomic techniques, the authors have shown the significance of fatty acid biosynthesis pathway behind the resistance mechanism. This study is significant as it sheds light on the role of an exogenous factor in altering membrane composition, polarization, and fluidity which ultimately leads to antimicrobial resistance.Strengths:(1) The experiments were carried out methodically and logically.(2) An adequate number of replicates were used for the experiments.Weaknesses:(1) The introduction section needs to be more informative and to the point.

Thank you so much for your suggestion. We have revised the introduction to make it more informative and to the point as following:

“Non-inheritable antibiotic or phenotypic resistance represents a serious challenge for treating bacterial infections. Phenotypic resistance does not involve genetic mutations Phenotypic resistance does not involve genetic mutations and is transient, allowing bacteria to resume normal growth. Biofilm and bacterial persisters are two phenotypic resistance types that have been extensively studied (Brandis et al., 2023; Corona & Martinez, 2013). Biofilms have complex structures, containing elements that impede antibiotic diffusion, sequestering and inhibiting their activity (Ciofu et al., 2022). Biofilm-forming bacteria and persisters also have distinct metabolic states that significantly reduce their antibiotic susceptibility (Yan & Bassler, 2019). These two types of phenotypic resistance share the common feature in their retarded or even cease of growth in the presence of antibiotics (Corona & Martinez, 2013). However, specific factors that promote phenotypic resistance and allow bacteria to proliferate in the presence of antibiotics remain poorly defined.

Metal ions have a diverse impact on the chemical, physical, and physiological processes of antibiotic resistance (Booth et al, 2011; Lu et al, 2020; Poole, 2017). This includes genetic elements that confer resistance to metals and antibiotics (Poole, 2017) and metal cations that directly hinder (or enhance) the activity of specific antibiotic drugs (Zhang et al., 2014). The metabolic environment can also impact the sensitivity of bacteria to antibiotics (Jiang et al., 2023; Lee & Collins, 2012; Peng et al., 2015; Zhang et al., 2020; Zhao et al., 2021). Light metal ions, such as magnesium, sodium, and potassium, can behave as cofactors for different enzymes (Du et al., 2016) and influence drug efficacy. Heavy metal ions, including Cu2+ and Zn2+, confer resistance to antibiotics (Yazdankhah et al., 2014; Zhang et al., 2018). Recent reports suggest that sodium negatively regulates redox states to promote the antibiotic resistance of Vibrio alginolyticus (Yang et al., 2018), while actively growing *Bacillus subtilis* cope with ribosome-targeting antibiotics by modulating ion flux (Lee et al, 2019). In Gram-negative bacteria, by contrast, zinc enhances antibiotic efficacy by potentiating carbapenem, fluoroquinolone, and β-lactam-mediated killing (Isaei et al., 2016; Zhang et al., 2014). Magnesium influences bacterial structure, cell motility, enzyme function, cell signaling, and pathogenesis (Wang et al., 2019). This mineral also modulates microbiota to harvest energy from the diet (Garcia-Legorreta et al., 2020), allowing *Bacillus subtilis* to cope with ribosome-targeting antibiotics by modulating ion flux (Lee et al., 2019). However, the role of magnesium in promoting phenotypic resistance is less well understood.

Vibrios inhabit seawater, estuaries, bays, and coastal waters, regions full of metal ions such as magnesium (Kumarage et al., 2022). Magnesium is the second most dissolved element in seawater after sodium. At a salinity of 3.5% seawater, the magnesium concentration is about 54 mM (Potis, 1968), and in deep seawater, can be as high as 2,500 mM (Wang et al., 2024). Vibrio parahaemolyticus and V. alginilyticus are two representative Vibrio pathogens that infect humans and aquatic animals, resulting in illness and economic loss, respectively (Grimes, 2020). (Fluoro)quinolones such as balofloxacin are used to treat Vibrio infection, however, resistance has emerged due to overuse (Suyamud et al., 2024). Indeed, (fluoro)quinolones are one of China's two primary residual chemicals associated with aquaculture (Liu et al., 2017). Vibrio can develop quinolone resistance through mutations in the DNA gyrase gene or through plasmid-mediated mechanisms (Dutta et al., 2021). Thus, the use of V. parahaemolyticus and V. alginilyticus as bacterial representatives, and balofloxacin as a quinolone-based antibacterial representative, can help to define novel magnesiumdependent phenotypic resistance mechanisms of pathogenic Vibrio species.

The current study evaluated whether magnesium induces phenotypic resistance in Vibrio species and defined the molecular/genetic basis for this resistance. Genetic approaches, GC-MS analysis of metabolite and membrane remodeling upon antibiotic exposure, membrane physiology, and extensive antimicrobial susceptibility testing were used for the evaluations.”

(2) The weakest point of this paper is in the logistics through the results section. The way authors represented the figures and interpreted them in the results section (or the figure legends) does not match. The figures are difficult to interpret and are not at all self-explanatory.

Thank you so much for your suggestion. We have followed your suggestion to check the match between result and figures. They are now revised.

(3) There are too many mislabeling of the figure panels in the main text which makes it difficult to find out which figures the authors are explaining. There should be more explanation on why and how they did the experiments and how the results were interpreted.

Thank you so much for your suggestion. We have checked the figures and main text to ensure that we make every figure clearly stated.

**Reviewer #2 (Public Review):**
Summary:In this study, the authors aimed to identify if and how magnesium affects the ability of two particular bacteria species to resist the action of antibiotics. In my view, the authors succeeded in their goals and presented a compelling study that will have important implications for the antibiotic resistance research community. Since metals like magnesium are present in all lab media compositions and are present in the host, the data presented in this study certainly will inspire additional research by the community. These could include research into whether other types of metals also induce multi-drug resistance, whether this phenomenon can be observed in other bacterial species, especially pathogenic species that cause clinical disease, and whether the underlying molecular determinants (i.e. enzymes) of metal-induced phenotypic resistance could be new antimicrobial drug targets themselves.Strengths:This study's strengths include that the authors used a variety of methodologies, all of which point to a clear effect of exogenous Mg2+ on drug resistance in the targeted species. I also commend the authors for carrying out a comprehensive study, spanning evaluation of whole cell phenotypes, metabolic pathways, genetic manipulation, to enzyme activity level evaluation. The fact that the authors uncovered a molecular mechanism underlying Mg2+-induced phenotypic resistance is particularly important as the key proteins should be studied further.Weaknesses:I believe there are weaknesses in the manuscript, however. The authors take for granted that the reader is familiar with all the assays utilized, and do not properly explain some experiments, and thus I highly suggest that the authors add a brief statement in each situation describing the rationale for each selected methodology (more details are in the private review to the authors). The Results section is also quite long and bogs down at times, and I suggest that the authors reduce its length by 10 to 20%. In contrast, the Introduction is sparse and lacks key aspects, for example, there should be mention of the study's main purpose and approaches, plus an introduction to the authors' choice of species and their known drug resistance properties, as well as the drug of choice (balofloxacin). Another notable weakness is that the authors evaluated Mg2+-induced phenotypic resistance only against two closely related species, and thus the generalizability of this mechanism of drug resistance is not known. The paper would be strengthened if the authors could demonstrate this type of phenotypic resistance in at least one more Gram-negative species and at least one Gram-positive species (antimicrobial susceptibility evaluations would suffice), each of which should be pathogenic to humans. Demonstrating magnesium-induced phenotypic drug resistance in the WHO Priority Bacterial Pathogens would be particularly important.In general, the conclusions drawn by the authors are justified by the data, except for the interpretation of some experiments. Importantly, this paper has discovered new antimicrobial resistance mechanisms and has also pointed to potential new targets for antimicrobials.

Thank you so much for your suggestion! We followed your idea the revise the manuscript as following:

(1) We added a brief statement in the situation to explain the result and methodology according to your suggestion in the private review.

(2) To make the streamline of the story more logic, we moved the whole second result to supplementary text and supplementary figure.

(3) We revised the introduction part by adding additional information to make it informative and to the point as following:

“Non-inheritable antibiotic or phenotypic resistance represents a serious challenge for treating bacterial infections. Phenotypic resistance does not involve genetic mutations Phenotypic resistance does not involve genetic mutations and is transient, allowing bacteria to resume normal growth. Biofilm and bacterial persisters are two phenotypic resistance types that have been extensively studied (Brandis et al., 2023; Corona & Martinez, 2013). Biofilms have complex structures, containing elements that impede antibiotic diffusion, sequestering and inhibiting their activity (Ciofu et al., 2022). Biofilm-forming bacteria and persisters also have distinct metabolic states that significantly reduce their antibiotic susceptibility (Yan & Bassler, 2019). These two types of phenotypic resistance share the common feature in their retarded or even cease of growth in the presence of antibiotics (Corona & Martinez, 2013). However, specific factors that promote phenotypic resistance and allow bacteria to proliferate in the presence of antibiotics remain poorly defined.

Metal ions have a diverse impact on the chemical, physical, and physiological processes of antibiotic resistance (Booth et al, 2011; Lu et al, 2020; Poole, 2017). This includes genetic elements that confer resistance to metals and antibiotics (Poole, 2017) and metal cations that directly hinder (or enhance) the activity of specific antibiotic drugs (Zhang et al., 2014). The metabolic environment can also impact the sensitivity of bacteria to antibiotics (Jiang et al., 2023; Lee & Collins, 2012; Peng et al., 2015; Zhang et al., 2020; Zhao et al., 2021). Light metal ions, such as magnesium, sodium, and potassium, can behave as cofactors for different enzymes (Du et al., 2016) and influence drug efficacy. Heavy metal ions, including Cu2+ and Zn2+, confer resistance to antibiotics (Yazdankhah et al., 2014; Zhang et al., 2018). Recent reports suggest that sodium negatively regulates redox states to promote the antibiotic resistance of Vibrio alginolyticus (Yang et al., 2018), while actively growing *Bacillus subtilis* cope with ribosome-targeting antibiotics by modulating ion flux (Lee et al, 2019). In Gram-negative bacteria, by contrast, zinc enhances antibiotic efficacy by potentiating carbapenem, fluoroquinolone, and β-lactam-mediated killing (Isaei et al., 2016; Zhang et al., 2014). Magnesium influences bacterial structure, cell motility, enzyme function, cell signaling, and pathogenesis (Wang et al., 2019). This mineral also modulates microbiota to harvest energy from the diet (Garcia-Legorreta et al., 2020), allowing *Bacillus subtilis* to cope with ribosome-targeting antibiotics by modulating ion flux (Lee et al., 2019). However, the role of magnesium in promoting phenotypic resistance is less well understood.

Vibrios inhabit seawater, estuaries, bays, and coastal waters, regions full of metal ions such as magnesium (Kumarage et al., 2022). Magnesium is the second most dissolved element in seawater after sodium. At a salinity of 3.5% seawater, the magnesium concentration is about 54 mM (Potis, 1968), and in deep seawater, can be as high as 2,500 mM (Wang et al., 2024). Vibrio parahaemolyticus and V. alginilyticus are two representative Vibrio pathogens that infect humans and aquatic animals, resulting in illness and economic loss, respectively (Grimes, 2020). (Fluoro)quinolones such as balofloxacin are used to treat Vibrio infection, however, resistance has emerged due to overuse (Suyamud et al., 2024). Indeed, (fluoro)quinolones are one of China's two primary residual chemicals associated with aquaculture (Liu et al., 2017). Vibrio can develop quinolone resistance through mutations in the DNA gyrase gene or through plasmid-mediated mechanisms (Dutta et al., 2021). Thus, the use of V. parahaemolyticus and V. alginilyticus as bacterial representatives, and balofloxacin as a quinolone-based antibacterial representative, can help to define novel magnesiumdependent phenotypic resistance mechanisms of pathogenic Vibrio species.

The current study evaluated whether magnesium induces phenotypic resistance in Vibrio species and defined the molecular/genetic basis for this resistance. Genetic approaches, GC-MS analysis of metabolite and membrane remodeling upon antibiotic exposure, membrane physiology, and extensive antimicrobial susceptibility testing were used for the evaluations.”

(4) We examined the effect of magnesium in WHO listed priority strains, which confirmed the results as following:

“Importantly, exogenous MgCl2 also increased MICs of clinic isolates, carbapenemresistant *Escherichia coli*, carbapenem-resistant *Klebsiella pneumoniae,* carbapenemresistant *Pseudomonas aeruginosa* and carbapenem-resistant *Acinetobacter baumannii* to balofloxacin (Fig 1G).”

**Recommendations for the authors:**

**Reviewer #1 (Recommendations For The Authors):**
(1) There are many grammatical mistakes to point out. The manuscript needs proofreading and editing.

We appreciate this comment! The manuscript has been revised by a native speaker.

(2) The introduction could be more informative. A little more description of magnesium - such as what it does to antibiotics and how it's known to affect the microbiome - might be helpful for the general readers. The question remains why out of all the metal ions that might affect antibiotic resistance (many of them are less explored), authors particularly decided to work on the effect of magnesium. The introduction should cover the rationale of their hypothesis. Also, the authors might want to briefly talk about the model organisms (V. algonolyticus and V. parahemolyticus) describing how threatening they are and how they are becoming resistant to antibiotics.

We appreciate this comment! We revise the introduction by providing additional information as following:

“In Gram-negative bacteria, by contrast, zinc enhances antibiotic efficacy by potentiating carbapenem, fluoroquinolone, and β-lactam-mediated killing (Isaei *et al.,* 2016; Zhang *et al.,* 2014). Magnesium influences bacterial structure, cell motility, enzyme function, cell signaling, and pathogenesis (Wang *et al.,* 2019). This mineral also modulates microbiota to harvest energy from the diet (Garcia-Legorreta *et al.,* 2020), allowing *Bacillus subtilis* to cope with ribosome-targeting antibiotics by modulating ion flux (Lee *et al.,* 2019). However, the role of magnesium in promoting phenotypic resistance is less well understood.

Vibrios inhabit seawater, estuaries, bays, and coastal waters, regions full of metal ions such as magnesium (Kumarage *et al.,* 2022). Magnesium is the second most dissolved element in seawater after sodium. At a salinity of 3.5% seawater, the magnesium concentration is about 54 mM (Potis, 1968), and in deep seawater, can be as high as 2,500 mM (Wang *et al.,* 2024). *Vibrio parahaemolyticus* and *V. alginilyticus* are two representative Vibrio pathogens that infect humans and aquatic animals, resulting in illness and economic loss, respectively (Grimes, 2020). (Fluoro)quinolones such as balofloxacin are used to treat Vibrio infection, however, resistance has emerged due to overuse (Suyamud *et al.,* 2024). Indeed, (fluoro)quinolones are one of China's two primary residual chemicals associated with aquaculture (Liu *et al.,* 2017). Vibrio can develop quinolone resistance through mutations in the DNA gyrase gene or through plasmid-mediated mechanisms (Dutta *et al.,* 2021). Thus, the use of *V. parahaemolyticus* and *V. alginilyticus* as bacterial representatives, and balofloxacin as a quinolone-based antibacterial representative, can help to define novel magnesiumdependent phenotypic resistance mechanisms of pathogenic Vibrio species.

The current study evaluated whether magnesium induces phenotypic resistance in Vibrio species and defined the molecular/genetic basis for this resistance. Genetic approaches, GC-MS analysis of metabolite and membrane remodeling upon antibiotic exposure, membrane physiology, and extensive antimicrobial susceptibility testing were used for the evaluations. ”

(3) Figure 1C is mislabeled as 1B (line 100). Line 101: The sentence is not clear and very confusing. What is meant by 15.6mM - 62.4 mM? Are they talking about the concentration of BLFX (though in the figure the concentration was shown in µg)? Please rewrite the sentence in a simplified way. Also, the zone of inhibition was decreased with increasing MgCl2, not increased.

We appreciate this comment! These have been revised, including that Fig 1B is now corrected as Fig. 1C. Line 101, which is now Line 122. The sentence was revised as following:

“At balofloxacin doses of 1.56, 3.125, 6.25, and 12.5 µg, the zone of inhibition decreased with increasing MgCl2 (Fig 1D)”

(4) In the western blot images, it would be nice to indicate the MW of the protein bands shown. The loading control used for the experiments should be clearly mentioned in the figure legends.

We appreciate this comment! The MWs are indicated in the western-blot image throughout the manuscript.

The loading control is clearly stated in the figure legend as following:

“Whole cell lysates resolved by SDS-PAGE gel was stained with Coomassie brilliant blue as loading control.”.

(5) Figures 2 B and C: the figure legend does not explain what the authors wanted to show. It's not clear how they plotted the inhibitory curve, or the binding efficacy. These panels need an explanation of how the analysis was done.

We appreciate this comment! The figure 2 is now removed to Suppl. Fig 2, and the description of figure 2 is moved to Suppl. Text. We revise the description of the result as following, which is in Suppl. Text:

“Prior studies suggest that the chelation of antibiotics by magnesium ions inhibits antibiotic uptake (Deitchman et al., 2018; Lunestad and Goksøyr, 1990). To investigate whether magnesium binds to balofloxacin, balofloxacin was pre-incubated with magnesium, and zone of inhibition (ZOI) analysis was conducted. Six different concentrations of balofloxacin were separately incubated with six different concentrations of MgCl2, and then spotted on filter paper so that a defined amount of balofloxacin could be used for ZOI. While lower concentrations of MgCl2, (0.78, 3.125, or 12.5 mM) did not alter the ZOI, higher concentrations, including 50 and 200 mM MgCl2, decreased the ZOI (Suppl. Fig 2A), suggesting that even high doses of magnesium had only a partial effect on balofloxacin through direct binding. For example, at 200 mM MgCl2 and 5 or 10 μg/mL balofloxacin, the balofloxacin ZOI was 53.2 and 70.3% of the ZOI at 0 mM MgCl2, suggesting that ≥50% of the antibiotics were still functional. Intracellular BLFX also decreased with increasing MgCl2 (Suppl. Fig 2B), while exogenous Mg2+ increased intracellular Mg2+ levels in a dose-dependent manner. For example, exogenous 50 and 200 mM MgCl2 increased intracellular Mg2+ levels to 1.21 and 1.31 mM, respectively (Suppl. Fig 2C). The relationship between TolC, an efflux pump that transports quinolones from bacterial cells, and Mg2+ was also assessed (Kobylka *et al.*, 2020; Song *et al.*, 2020). The expression of TolC/*tolC* was unaffected by Mg2+ (Suppl. Fig 2D). Magnesium is critical for LPS stability. LPS levels increased at 200 mM Mg2+ (Suppl. Fig 2E), however, the loss of *waaF, lpxA*, and *lpxC*, three key genes involved in LPS biosynthesis, did not influence balofloxacin sensitivity/resistance in the presence of Mg2+ (Suppl. Fig 2F). These findings suggest that magnesium-induced LPS biosynthesis does not contribute directly to BLFX resistance and demonstrate that Mg2+ influx is involved in balofloxacin resistance.”

(6) For the metabolomics results, it will help immensely if the authors provide a volcano plot of the identified metabolites and plot the heat map according to the -log2 metabolite intensities. In Figure 3A, it's not clear what information is conveyed through Euclidean distance calculations of the heat map. In Figure 3 B, the authors mentioned that the OPLS-DA test was conducted, although the figure shows a PCA plot, so it's not clear how these two are connected. Figure 3 E: the figure legend says scattered plot, but the panel represents color-coded numerical values, not a scattered plot. Also, it's not clear how they got those values.

We appreciate this comment! We quite agree with you that if the differential metabolites could be shown as volcano plot. However, we didn’t adopt volcano plot in this study because this is a magnesium concentration-dependent metabolomes that includes 6 groups in parallel. Volcano plots may give a complex view of the comparison among different groups. We also tried to plot the heat map according to the -log2 metabolite intensities. Although this analysis cluster 200 mM and 50 mM groups better, the data of low magnesium concentrations was not consistent, which may be due to the minor metabolic change of low concentrations magnesium. Thank you for your understanding.

For Euclidean distance calculations, we explain in the figure legend as following:

“Euclidean distance calculations were used to generate a heatmap that shows clustering of the biological and technical replicates of each treatment.”

In Figure 2B, which was Figure 3B in previous version, it has been replaced with OPLS-DA analysis in the revised version.

In Figure 2E, which was Figure 3E in previous version, it is revised as following:

“E. Areas of the peaks of palmitic acid and stearic acid generated by GC-MS analysis.”

(7) In Figure 4, the figure legends (as well as the in the text) are not properly referred to. Please make sure to refer to the correct panel.

We appreciate this comment! The figure legends have been corrected to match the panel and text.

Figure 4F: how was the synergy analysis done? In the methods section, the authors described the antibiotic bactericidal assay protocol, but there was no clear indication of how they generated the isobologram.

We appreciate this comment! We provide additional information in the Figure 3F legend, which was Figure 4F in previous version, as following:

“Synergy analysis for BFLX with palmitic acid for *V. alginolyticus*. Synergy was performed by comparing the dose needed for 50% inhibition of the synergistic agents (white) and non-synergistic (i.e., additive) agents (purple).”

(8) Figure 5 A: the scatter plot is plotted according to the area along the Y axis: which "area" is represented here? There is absolutely no explanation, neither in the results nor in the figure legends. Using box plots might be a better option than using a scattered plot.

We appreciate this comment! “Area” has been noted in the revised manuscript as following:

“The area indicates the area of the peak of the metabolite in total ion chromatography of GC-MS.”

(9) In Figure 6 A, the heat map is plotted according to the column Z scores. What is meant by "column Z score"? The corresponding figure legend says, "heat map showing differential abundance of lipid". Z scores do not represent an abundance of a variable, so the conclusion might not be appropriate here.

We appreciate this comment! In Figure 5A, which was Figure 6A in previous version, column Z score shows the abundance of metabolites analyzed, which is automatically generated in the heat map analysis to give a sign of these metabolites tested. The legend has been revised as following:

“Heatmap showing changes in differential lipid levels at the indicated concentration of MgCl2.”

(10) Line 313-314: it should be Figure EV6C.

We appreciate this comment! The citation has been corrected.

(11) The authors have shown that Mg+2 does not alter the LPS transport system, however, there was some significant increase in LPS expression at 200mM MgCl2. It would be interesting if the authors could also check if Mg+2 has any effect on the outer membrane protein (OMP) integrity (by checking OMP components BamA and LptD).

We appreciate this comment! We have carefully examined the membrane permeability in Figure 7. We thus didn’t perform additional experiment here to see the change of BamA and LptD. Thank you very much for your understanding.

(12) I wonder if the authors could check the effect of extracellular Mg+2 during the co-treatment of palmitic acid, linoleic acid, and balofloxacin. Will there still be the antagonistic effect or the presence of Mg+2 could change the phenotype?

We appreciate this comment! Additional experiments is performed as following:

“Furthermore, magnesium had a minimal effect on the antagonistic effect of palmitic acid, linolenic acid, and balofloxacin (Fig 4G), suggesting that this mineral functions through lipid metabolism.”

**Reviewer #2 (Recommendations For The Authors)**:(1) As mentioned in the Public Review, I strongly believe that the impact of this study will be more significant if magnesium-induced phenotypic drug resistance could be demonstrated in at least one other Gram-negative and one other Grampositive species, both of which should be human pathogens. The full suite of experiments would not be necessary for this suggestion; evaluation of the effect of Mg concentration in growth media on the drug resistance of other species, testing the different antibiotic types used in this study, would be sufficient.

We appreciate this comment! Additional experiments have performed to test this idea. Mg2+ has the similar effect on carbapenem-resistant *Escherichia coli*, carbapenem-resistant *Klebsiella pneumoniae,* carbapenem-resistant *Pseudomonas aeruginosa* and carbapenem-resistant *Acinetobacter baumannii* as the similar as on the Vibrio species in shown in Figure 1G. These have been described following as

“Importantly, exogenous MgCl2 also increased MICs of clinic isolates, carbapenemresistant *Escherichia coli*, carbapenem-resistant *Klebsiella pneumoniae,* carbapenemresistant *Pseudomonas aeruginosa* and carbapenem-resistant *Acinetobacter baumannii* to balofloxacin (Fig 1G).”

(2) I recommend that the Introduction section be expanded. I recommend one or two sentences introducing the two Vibrio species selected for study. I.e. why did the authors choose these two species? What is known about their phenotypic drug resistance in the literature? Why did the authors select balofloxacin for their studies, is it a common antimicrobial used vs Vibrios? As well, the end of the Introduction section ends abruptly with no transition to the present study itself. The end of the introduction should include one or two sentences introducing the main purpose of the study, its approach, and the techniques undertaken. For example, "In this study, we evaluated whether magnesium induces phenotypic resistance in Vibrio species and the molecular/genetic basis for such resistance. We used genetic approaches, GC-MS analysis of metabolite and membrane remodeling upon antibiotic exposure, membrane physiology, and extensive antimicrobial susceptibility evaluations."

We appreciate this comment! We revise the introduction by providing additional information as following:

“In Gram-negative bacteria, by contrast, zinc enhances antibiotic efficacy by potentiating carbapenem, fluoroquinolone, and β-lactam-mediated killing (Isaei *et al.,* 2016; Zhang *et al.,* 2014). Magnesium influences bacterial structure, cell motility, enzyme function, cell signaling, and pathogenesis (Wang *et al.,* 2019). This mineral also modulates microbiota to harvest energy from the diet (Garcia-Legorreta *et al.,* 2020), allowing *Bacillus subtilis* to cope with ribosome-targeting antibiotics by modulating ion flux (Lee *et al.,* 2019). However, the role of magnesium in promoting phenotypic resistance is less well understood.

Vibrios inhabit seawater, estuaries, bays, and coastal waters, regions full of metal ions such as magnesium (Kumarage *et al.,* 2022). Magnesium is the second most dissolved element in seawater after sodium. At a salinity of 3.5% seawater, the magnesium concentration is about 54 mM (Potis, 1968), and in deep seawater, can be as high as 2,500 mM (Wang *et al.,* 2024). *Vibrio parahaemolyticus* and *V. alginilyticus* are two representative Vibrio pathogens that infect humans and aquatic animals, resulting in illness and economic loss, respectively (Grimes, 2020). (Fluoro)quinolones such as balofloxacin are used to treat Vibrio infection, however, resistance has emerged due to overuse (Suyamud *et al.,* 2024). Indeed, (fluoro)quinolones are one of China's two primary residual chemicals associated with aquaculture (Liu *et al.,* 2017). Vibrio can develop quinolone resistance through mutations in the DNA gyrase gene or through plasmid-mediated mechanisms (Dutta *et al.,* 2021). Thus, the use of *V. parahaemolyticus* and *V. alginilyticus* as bacterial representatives, and balofloxacin as a quinolone-based antibacterial representative, can help to define novel magnesiumdependent phenotypic resistance mechanisms of pathogenic Vibrio species.

The current study evaluated whether magnesium induces phenotypic resistance in Vibrio species and defined the molecular/genetic basis for this resistance. Genetic approaches, GC-MS analysis of metabolite and membrane remodeling upon antibiotic exposure, membrane physiology, and extensive antimicrobial susceptibility testing were used for the evaluations. ”

(3) The authors introduce the acronym AWST but never use it again in the paper, instead they use SWT. The authors should introduce SWT only for consistency.

We appreciate this comment! We have corrected all the “SWT” to “ASWT”

(4) Line 76 is not clear: what is meant by "some of which could influence drug efficacy" - the enzymes that utilize light metal ions are co-factors? Or the metals directly?

We appreciate this comment! The information we wanted to deliver is that light metal ions can serve as cofactors to catalyze biochemical reaction. Such chemical reaction would alter the drug efficacy, e.g. the Fe-S cluster are metallocofactor for proteins which regulates redox chemistry including antibioticinduced redox change. However, this information is not appropriate for this manuscript, so we delete this sentence.

(5) Line 90: add a reference corroborating that this chemical composition is a mimic of marine water. The NaCl concentration used in particular looks quite low.

We appreciate this comment! It was a typo error. The NaCl concentration was 210 mM as shown in Suppl. Table 1. We also provide details of the chemical composition of the marine water as following:

“Marine environments and agriculture, where antibiotics are commonly used, are rich in magnesium. To investigate whether this mineral impacts antibiotic activity, the minimal inhibitory concentration (MIC) of *V. alginolyticus* ATCC33787 and *V. parahaemolyticus* VP01, which we referred as ATCC33787 and VP01 afterwards, isolated from marine aquaculture, to balofloxacin (BLFX) in Luria-Bertani medium

(LB medium) plus 3% NaCl as LBS medium and “artificial seawater” (ASWT) medium that included the major ion species in marine water (Wilson, 1975) (LB medium plus 210 mM NaCl, 35 mM Mg2SO4, 7 mM KCl, and 7 mM CaCl2) were assessed”

(6) Line 98 and Figure 1B. M9 is indicated in the text but does not appear in the figure, the figure only shows SWT. This should be checked. Line 99: based on Figure 1C, the authors are adding MgCl2 to SWT, SWT should be mentioned in this line. Line 100: I believe this is referring to Figure 1C, which should be checked.

We appreciate this comment!

Line 98, which is now Line 118: We have corrected M9 to ASWT as following:

“However, the MIC for BLFX was higher in ASWT medium supplemented with Mg2SO4 or MgCl2 than in LB medium (Fig 1B).”

Line 99, which is now Line 133: the sentence is corrected as following:

“The MIC for BLFX increased at higher concentrations of MgCl2 in ASWT”

Line 100, which is now Line 135: we have corrected Fig 1B to Fig. 1C.

(7) Line 101: text and Figure 1D are not consistent, as Figure 1D does not show this level of precision in added MgCl2 as indicated in the text (15.6 - 62.4 mM).

We appreciate this comment! The sentence has been corrected as following: “At balofloxacin doses of 1.56, 3.125, 6.25, and 12.5 µg, the zone of inhibition decreased with increasing MgCl2 (Fig 1D)””.

(8) MgCl2 clearly induces increasing levels of BLFX resistance, and to high levels, but not for every antibiotic. For example, the level of increased resistance to blactams is low (ceftriaxone) and plateaus (ceftazidime). As well, resistance to gentamicin plateaus at a lower level than the other aminoglycosides. These observations do not take away from the conclusion that Mg induces multi-drug resistance, but since the behaviour of the MICs for these drugs is different than the other drugs, they should be mentioned. Also, Figure 1F - tetracyclines (plural) is used for vertical axis label - does this refer to the tetracycline itself or the class itself, and if the class, which one was tested?

We appreciate this comment! We revise the description as following: “Notably, magnesium had a reduced effect on ceftriaxone and gentamicin than other antibiotics.”

The tetracyclines is labeled as “Oxytetracycline” in the revised manuscript.

- The magnesium chelation experiments presented in Figure 2 are not clear. The authors should briefly mention how this was done around line 128, and what data underlies the values in Figure 2C. Figure 2B is also not clear to me at all. Similarly, how the authors measured intracellular balofloxacin and Mg2+ is not clear and should be mentioned briefly around lines 130-132.

We appreciate this comment! These have been rewritten following as “To investigate whether magnesium binds to balofloxacin, balofloxacin was preincubated with magnesium, and zone of inhibition (ZOI) analysis was conducted. Six different concentrations of balofloxacin were separately incubated with six different concentrations of MgCl2, and then spotted on filter paper so that a defined amount of balofloxacin could be used for ZOI. While lower concentrations of MgCl2, (0.78, 3.125, or 12.5 mM) did not alter the ZOI, higher concentrations, including 50 and 200 mM MgCl2, decreased the ZOI (Suppl. Fig 2A), suggesting that even high doses of magnesium had only a partial effect on balofloxacin through direct binding. For example, at 200 mM MgCl2 and 5 or 10 μg/mL balofloxacin, the balofloxacin ZOI was 53.2 and 70.3% of the ZOI at 0 mM MgCl2, suggesting that ≥50% of the antibiotics were still functional. Intracellular BLFX also decreased with increasing MgCl2 (Suppl. Fig 2B), while exogenous Mg2+ increased intracellular Mg2+ levels in a dose-dependent manner. For example, exogenous 50 and 200 mM MgCl2 increased intracellular Mg2+ levels to 1.21 and 1.31 mM, respectively (Suppl. Fig 2C). The relationship between TolC, an efflux pump that transports quinolones from bacterial cells, and Mg2+ was also assessed (Kobylka *et al.*, 2020; Song *et al.*, 2020). The expression of TolC/*tolC* was unaffected by Mg2+ (Suppl. Fig 2D). Magnesium is critical for LPS stability. LPS levels increased at 200 mM Mg2+ (Suppl. Fig 2E), however, the loss of *waaF, lpxA*, and *lpxC*, three key genes involved in LPS biosynthesis, did not influence balofloxacin sensitivity/resistance in the presence of Mg2+ (Suppl. Fig 2F). These findings suggest that magnesium-induced LPS biosynthesis does not contribute directly to BLFX resistance and demonstrate that Mg2+ influx is involved in balofloxacin resistance.”

- Line 135: LPS cannot be "expressed", as the authors word it here. This should be corrected. Also, the inspection of Figure 2G actually shows the levels of LPS increase with increased Mg2+. The authors should re-evaluate these results and change their description around this area of the Results.

We appreciate this comment! We have removed the whole Figure 2 to Supplementary Text and Supplementary Figure 2. We rewrite this part as following: “The relationship between TolC, an efflux pump that transports quinolones from bacterial cells, and Mg2+ was also assessed (Kobylka *et al.*, 2020; Song *et al.*, 2020). The expression of TolC/*tolC* was unaffected by Mg2+ (Suppl. Fig 2D). Magnesium is critical for LPS stability. LPS levels increased at 200 mM Mg2+ (Suppl. Fig 2E), however, the loss of *waaF, lpxA*, and *lpxC*, three key genes involved in LPS biosynthesis, did not influence balofloxacin sensitivity/resistance in the presence of Mg2+ (Suppl. Fig 2F). These findings suggest that magnesium-induced LPS biosynthesis does not contribute directly to BLFX resistance and demonstrate that Mg2+ influx is involved in balofloxacin resistance.”

- Section: MgCl2 affects bacterial metabolism. Authors switched to M9 medium - why? This contrasts with other sections using SWT and should be explained. Also, I cannot evaluate whether the statistical analysis of the data here was performed correctly and was appropriate for this type of experiment. I advise the authors to move the details in lines 166-169 to the Materials and Methods and replace this section instead with a more accessible description of the statistical analysis that a non-expert would be able to appreciate. Furthermore, analysis of Figure 3A indicates that the levels of asparagine, 4-hydroxybutyric acid, uracil, cystathionine, fumaric acid, and aminoethanol have significantly changed at high MgCl2, but these are not mentioned in the text. I suggest the authors mention these if they are relevant to the 12 enriched pathways, especially the biosynthesis of fatty acids.

We appreciate this comment!

We indicate the reason we use M9 medium as following:

“To better understand how magnesium affects bacterial metabolism” for explaining why the M9 medium was used.”

The information lines 166-169 indicated has been removed to M &M.

We have carefully examined the abundance of the metabolites and the enriched pathway. Among the listed metabolites, only fumarate is within the enriched pathways. We mention this point in our revised manuscript as following:

“The increase in fatty acid biosynthesis could be partially explained by an imbalanced pyruvate cycle/TCA cycle, in which fumarate levels increased at higher Mg2+ while succinate levels increased at lower Mg2+ (Suppl. Fig 5B). These findings indicated that glycolysis fluxes into fatty acid biosynthesis rather than the pyruvate cycle/TCA cycle. The relevance of fatty acids and BLFX was demonstrated by the observation that exogenous palmitic acid increased bacterial resistance to balofloxacin (Fig 2F). These results suggest that fatty acid metabolism may be critical to magnesium-based phenotypic resistance.”

- Line 211 appears to refer to Figure 4F and should be checked. Similarly in line 216 - appears this should be Figure 4H, and line 218 should be Figure 4H. Line 226: add a reference to Fig 4I (after arcA was decreased). Line 227: what are genes N646_1004 and N646_1885? Based on Fig 4J these are crp - authors should add to line 227. Line 228 appears to refer to Figure 4J, not Figure 4I. Line 229 - should be Figure 4K, not Figure 4I. Line 231 - should be 4L, not 4K. Line 239 - should be 4M.

We appreciate this comment! The text and figure is now matched.

- Line 312: the descriptions of "11 lipids, 32 lipids, and 53", and then "26 lipids, 52 lipids, and 107 lipids" are not clear at all and should be corrected.

We appreciate this comment! The sentence is revised as following:

“The abundance of 11, 32, and 53 lipids was increased in 3.125, 50, and 200 mM MgCl2-treated bacteria, respectively, while the abundance of 26, 52, and 107 lipids was decreased in 3.125, 50, and 200 mM MgCl2-treated bacteria, respectively (Suppl. Fig 7C)”

- Line 340. What is the assay the authors are using to measure the levels of the PGS and PSS enzymes? This is not mentioned or clear in this part of the Results.

We appreciate this comment! We provide the information in the manuscript as following:

“Levels of PGS and PSS were quantified by ELISA kits according to manufacture’s instruction (Shanghai Fusheng Industrial Co., Ltd., China)”

- Line 372: What is the assay for measuring membrane depolarization? This is not mentioned and I suggest it should be. Line 374: Figure 7B does not show time dependence, only dose dependence, this should be corrected, it is assumed the authors are referring to Fig 7C for the time dependence data.

We appreciate this comment! We provide the information in the result as following:

“The voltage-sensitive dye, DiBAC4(3) showed that 12.5–200 mM MgCl2 promoted membrane depolarization in a dose-dependent manner (Fig 6A)”

We also explain how DiBAC4(3) can be used to measure membrane depolarization in the Materials and Methods section as following:

“DiBAC4(3) is a s voltage-sensitive probe that penetrates depolarized cells, binding intracellular proteins or membranes exhibiting enhanced fluorescence and red spectral shift.”

To make it clear the specific figure, we revise the sentence as following:

“Meanwhile, MgCl2 had a dose-dependent (Fig 6B) and time-dependent (Fig 6C) effect on proton motive force (PMF).”

- Line 384: mention how FM5-95 measures membrane permeability. The authors should also clarify how this reagent is used to measure membrane fluidity, and it is not clear if the data for this is presented in Figure 7 - please clarify. Regarding SYTO9 dye experiment: the authors should briefly explain the experimental design - how SYTO9 dye operates and why FACS was chosen. What is labeled with FITC?

We appreciate this comment! We clarify the reason we use FM5-95 in the Methods and Materials section as following:

“Measurement of fluidity by fluorescence microscopy

Measurement of membrane fluidity is performed as previously described (Wen et al., 2022). Briefly, ATCC33787 were cultured in medium with indicated concentrations of MgCl2, collected and then adjusted to OD 0.6. Aliquot of 100 μL bacteria cells of each sample were diluted to 1 mL and 10 μL (10 mg/mL) FM5-95 (Thermo Fisher Scientific, USA) was added. FM5-95 is a lipophilic styryl dye that insert into the outer leaflet of bacterial membrane and become fluorescence. This dye preferentially bind to the microdomains with high membrane fluidity(Wen et al., 2022). After incubated for 20 min at 30 ℃ at vibration without light, the sample was centrifuged for 10 min at 12,000 rpm. The pellets were resuspended with 20 μL of 3% NaCI. Aliquot of 2 μL sample was dropped on the agarose slide, and take photos under the inverted fluorescence microscope.”

This data is presented as micrographs in Fig. 6D, which shows the decreased FM5-95 staining with increasing concentrations of MgCl2. We make this description clear with the following revision:

“FM5-95 staining decreased with increasing concentrations of Mg2+, and no staining was observed in the presence of 200 mM Mg2+ (Fig 6D).”

We explain the reason why we use SYTO9 as following:

“SYTO9, a green fluorescent dye that binds to nucleic acid, enters and stains bacteria cells when there is an increase in membrane permeability (Lehtinen *et al.,* 2004; McGoverin *et al.,* 2020). Staining decreased with increasing MgCl2, indicating that bacterial membrane permeability declined in an Mg2+ dose-dependent manner (Fig 6E).”

We didn’t use FACS in this study, while we only analyze the fluorescence distribution with the equipment. To make it clear, we revise the sentence as following:

“After incubated for 15 min at 30 ℃ at vibration without light, the mixtures were filtered and measured by flow cytometry (BD FACSCalibur, USA).”

- Lines 391-397. The statement that palmitic acid shifts the peaks in Figure 7F is not supported by the data. There is essential no change in the major peak position within each MgCl2 concentration set with increasing palmitic acid. For the linolenic acid data, it is clear that linolenic acid increases permeability only at 50 mM MgCl2-this should be mentioned in the text.

We appreciate this comment! We revise the sentence as following:

“Exogenous palmitic acid also shifted the fluorescence signal peaks to the left in an MgCl2-dependent manner while palmitic acid only slightly shifted the peaks (Fig 6F). In contrast, exogenous linolenic acid shifted the peak to the right in a dose-dependent manner at 50 mM MgCl2 (Fig 6G).”

- Line 404-405 - as mentioned earlier, the assay for the update of BLFX should be mentioned (if it is done so earlier in the text, then it does not need to be here).

We appreciate this comment! It has been mentioned in the introduction.

- Discussion: CpxA/R-OmprF pathway is mentioned here for the first time. Is this one of the pathways modified by MgCl2 as determined during the course of the study? If so, this should be reworded to mention that. If not, the relevance of this particular pathway as it relates to light metals and phenotypic resistance should be discussed.

We appreciate this comment! Since it is not relevant to the discussion of Mg2+ and fatty acid biosynthesis, we delete this sentence in the revised manuscript.

-The following grammatical errors should be corrected:-line 55 change to: "genetic mutations; instead, this type of resistance is transient, and bacteria resume normal growth"-line 57: change to "resistance types are biofilm"-line 61: change to "states that significantly"-line 63: change to "resistance share the common feature in they retard or even cease in the presence"-line 65: change to "resistance that allow bacteria to proliferate"-line 81: change "But whether" to "Whether"-line 178: change to "may be critical to the Mg-based phenotypic resistance"-line 86: change to "Marine environments and agriculture are rich in magnesium, where..."-line 93: change in to vs-line 154: insert space after metabolism-line 158: change 'identified" to "focused on the levels of"-line 160: change "The levels of forty-one metabolites"-line 198: change shared to share-line 310: increased is duplicated, delete one-line 451: add "the" before ratio-line 453: gram should be capitalized-line 462: "the regulation" should be reworded to "More importantly, the effect of exogenous MgCl targets the..."-line 469: add dash between Mg2+ and limited-line 478: change "the crucial" to "a crucial"-there are numerous locations in the manuscript where the word "magnetism" is used when clearly the word is supposed to be magnesium - this should be corrected

We appreciate this comment! These have been corrected or revised.

**Editors comments:**
Page 2 line 27; Page 25 line number 426; page 27 line number 481: In the abstract and discussion, only *Vibrio alginolyticus* was mentioned, even though two Vibrio species were used in the study. It would be helpful to understand the rationale behind the focus on this particular species.

We appreciate this comment! We have revised the introduction to provide additional information as following:

“Vibrios inhabit seawater, estuaries, bays, and coastal waters, regions full of metal ions such as magnesium (Kumarage *et al.,* 2022). Magnesium is the second most dissolved element in seawater after sodium. At a salinity of 3.5% seawater, the magnesium concentration is about 54 mM (Potis, 1968), and in deep seawater, can be as high as 2,500 mM (Wang *et al.,* 2024). *Vibrio parahaemolyticus* and *V. alginilyticus* are two representative Vibrio pathogens that infect humans and aquatic animals, resulting in illness and economic loss, respectively (Grimes, 2020). (Fluoro)quinolones such as balofloxacin are used to treat Vibrio infection, however, resistance has emerged due to overuse (Suyamud *et al.,* 2024). Indeed, (fluoro)quinolones are one of China's two primary residual chemicals associated with aquaculture (Liu *et al.,* 2017). Vibrio can develop quinolone resistance through mutations in the DNA gyrase gene or through plasmid-mediated mechanisms (Dutta *et al.,* 2021). Thus, the use of *V. parahaemolyticus* and *V. alginilyticus* as bacterial representatives, and balofloxacin as a quinolone-based antibacterial representative, can help to define novel magnesium-dependent phenotypic resistance mechanisms of pathogenic Vibrio species.”

On Page 2, line 34: The abstract contains some undefined abbreviations, such as 'PE' and 'PG', which should be explained.

We appreciate this comment! We explain the PE and PG in the revised abstract as following:

“phosphatidylethanolamine (PE) biosynthesis is reduced and phosphatidylglycerol (PG)”

On Page 2, line 31-32: For the statement "Exogenous supplementation of fatty acids confirm the role of fatty acids in antibiotic resistance…" it would be beneficial to specify whether the fatty acids were saturated or unsaturated.

Response, We appreciate this comment! We revise the sentence as following:

“Exogenous supplementation of unsaturated and saturated fatty acids increased and decreased bacterial susceptibility to antibiotics, respectively, confirming the role of fatty acids in antibiotic resistance.”

The potential effects of the specific ions (SO4 and Cl2) present in the Mg2SO4 and MgCl2 compounds used in the study were not discussed. It would be useful to understand if these ions had any influence on the observed outcomes.

We appreciate this comment! We revise the sentence as following:

“However, the MIC for BLFX was higher in ASWT medium supplemented with Mg2SO4 or MgCl2 than in LB medium (Fig 1B). And Mg2SO4 or MgCl2 had no

difference on MIC, suggesting it is Mg2+ not other ions contribute to the MIC change.”

On Page 8, line 141: The heading of Figure 2, "Mg2+ elevates intracellular Mg2+," seems redundant and could be revised for clarity or modified.

We appreciate this comment! Figure 2 is now moved to supplementary figure as Suppl. Fig 2. The title is revised as following:

“Figure 2. Mg2+ decreases balofloxacin uptake.”

On Page 4, line 91: some terms/abbreviations, such as 'LB' and 'M9,' require expansion or definition to ensure the reader's understanding.

We appreciate this comment! We include the expansion for LB and M9 in the revised manuscript as following:

“Luria-Bertani medium (LB medium)” and “M9 minimal medium (M9 medium)”

Page 4, line 92: The real seawater composition used in the experiments should be supported by a reference.

We appreciate this comment! We provide the reference in the revised manuscript as following:

““artificial seawater” (ASWT) medium that included the major ion species in marine water (Wilson, 1975) (LB medium plus 210 mM NaCl, 35 mM Mg2SO4, 7 mM KCl, and 7 mM CaCl2)”

Page 4 line, number 93: the he full names of the bacterial strains (e.g., ATCC33787 and VP01) should be provided instead of just the strain numbers.

We appreciate this comment! We revised the sentence as following:

“To investigate whether this mineral impacts antibiotic activity, the minimal inhibitory concentration (MIC) of *V. alginolyticus* ATCC33787 and *V. parahaemolyticus* VP01, which we referred as ATCC33787 and VP01 afterwards,”

Finally, there appears to be a potential contradiction between the statements on page 12, lines 211-212 and 214-216, regarding the effects of Mg2+ on the synthesis of unsaturated fatty acids. Further explanation may be needed to reconcile these seemingly contradictory points.

We appreciate this comment! For line 221-226, which was previously line 211-212, is about the gene expression for fatty acid biosynthesis. While, Line 228 and 233, which was previously line 214-216 is about the gene expression for fatty acid degradation. We agree that the previous description is a little bit confuse. We revise the sentence to emphasize that we focus on fatty acid degradation so that the readers can tell them apart.

In the text, we revised it as following:

“In addition, we also quantified gene expression during fatty acid degradation to determine whether Mg2+ affects this process” In the figure legend, we also indicate that

“H. qRT-PCR for the expression of genes encoding fatty acid degradation in the absence or presence of the indicated concentrations of MgCl2”